# LLM-grounded Video Diffusion Models

**Long Lian**[1*]**, Baifeng Shi**[1*]**, Adam Yala**[1,2†]**, Trevor Darrell**[1†]**, Boyi Li**[1†]
[1]UC Berkeley   [2]UCSF
{longlian,baifeng_shi,yala,trevordarrell,boyili}@berkeley.edu

## Abstract

Text-conditioned diffusion models have emerged as a promising tool for neural video generation. However, current models still struggle with intricate spatiotemporal prompts and often generate restricted or incorrect motion. To address these limitations, we introduce LLM-grounded Video Diffusion (LVD). Instead of directly generating videos from the text inputs, LVD first leverages a large language model (LLM) to generate dynamic scene layouts based on the text inputs and subsequently uses the generated layouts to guide a diffusion model for video generation. We show that LLMs are able to understand complex spatiotemporal dynamics from text alone and generate layouts that align closely with both the prompts and the object motion patterns typically observed in the real world. We then propose to guide video diffusion models with these layouts by adjusting the attention maps. Our approach is training-free and can be integrated into any video diffusion model that admits classifier guidance. Our results demonstrate that LVD significantly outperforms its base video diffusion model and several strong baseline methods in faithfully generating videos with the desired attributes and motion patterns.

## 1 Introduction

Text-to-image generation has made significant progress in recent years (Saharia et al., 2022; Ramesh et al., 2022). In particular, diffusion models (Ho et al., 2020; Dhariwal & Nichol, 2021; Ho et al., 2022b; Nichol et al., 2021; Nichol & Dhariwal, 2021; Rombach et al., 2022) have demonstrated their impressive ability to generate high-quality visual contents. Text-to-video generation, however, is more challenging, due to the complexities associated with intricate spatial-temporal dynamics. Recent works (Singer et al., 2022; Blattmann et al., 2023; Khachatryan et al., 2023; Wang et al.,

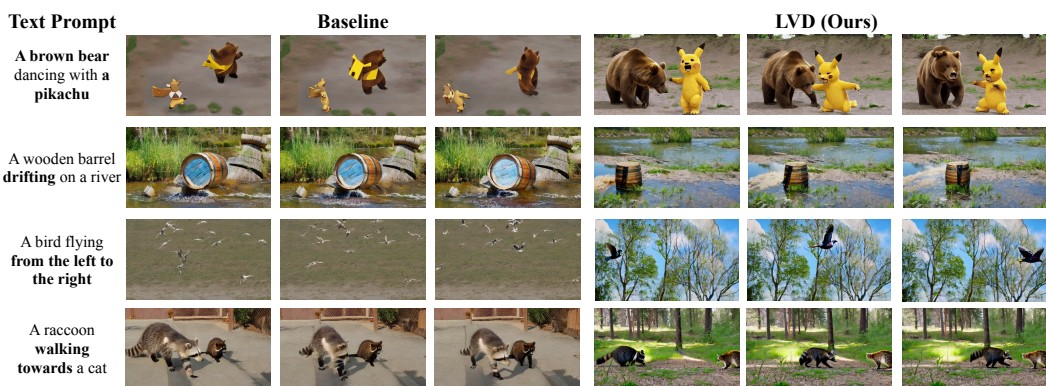

Figure 1: *Left:* Existing text-to-video diffusion models such as Wang et al. (2023) often encounter challenges in generating high-quality videos that align with complex prompts. *Right:* Our training-free method LVD, when *applied on the same model*, allows the generation of realistic videos that closely align with the input text prompt.

---

[*]Equal contribution.
[†]Equal advising.

2023) have proposed text-to-video models that specifically aim to capture spatiotemporal dynamics in the input text prompts. However, these methods still struggle to produce realistic spatial layouts or temporal dynamics that align well with the provided prompts, as illustrated in Fig. 1.

Despite the enormous challenge for a diffusion model to generate complex dynamics directly from text prompts, one possible workaround is to first generate explicit spatiotemporal layouts from the prompts and then use the layouts to control the diffusion model. In fact, recent work (Lian et al., 2023; Feng et al., 2023; Phung et al., 2023) on text-to-image generation proposes to use Large Language Models (LLMs) (Wei et al., 2022; OpenAI, 2020; 2023) to generate spatial arrangement and use it to condition text-to-image models. These studies demonstrate that LLMs have the surprising capability of generating detailed and accurate coordinates of spatial bounding boxes of each object based on the text prompt, and the bounding boxes can then be utilized to control diffusion models, enhancing the generation of images with coherent spatial relationships. However, it has not been demonstrated if LLMs can generate *dynamic* scene layouts for videos on both spatial and temporal dimensions. The generation of such layouts is a much harder problem, since an object's motion patterns often depend on both the physical properties (*e.g.*, gravity) and the object's attributes (*e.g.*, elasticity vs rigidity).

In this paper, we investigate whether LLMs can generate spatiotemporal bounding boxes that are coherent with a given text prompt. Once generated, these box sequences, termed Dynamic Scene Layouts (DSLs), can serve as an intermediate representation bridging the gap between the text prompt and the video. With a simple yet effective attention-guidance algorithm, these LLM-generated DSLs are leveraged to control the generation of object-level spatial relations and temporal dynamics in a training-free manner. The whole method, referred to as LLM-grounded Video Diffusion (LVD), is illustrated in Fig. 2. As shown in Fig. 1, LVD generates videos with the specified temporal dynamics, object attributes, and spatial relationships, thereby substantially enhancing the alignment between the input prompt and the generated content.

To evaluate LVD's ability to generate spatial layouts and temporal dynamics that align with the prompts, we propose a benchmark with five tasks, each requiring the understanding and generation of different spatial and temporal properties in the prompts. We show that LVD significantly improves the text-video alignment compared to several strong baseline models. We also evaluate LVD on common datasets such as UCF-101 (Soomro et al., 2012) and MSR-VTT (Xu et al., 2016) and conducted an evaluator-based assessment, where LVD shows consistent improvements over the base diffusion model that it uses under the hood.

**Contributions. 1)** We show that text-only LLMs are able to generate dynamic scene layouts that generalize to previously unseen spatiotemporal sequences. **2)** We propose LLM-grounded Video Diffusion (LVD), the first training-free pipeline that leverages LLM-generated dynamic scene layouts for enhanced ability to generate videos from intricate text prompts. **3)** We introduce a benchmark for evaluating the alignment between input prompts and the videos generated by text-to-video models.

## 2 RELATED WORK

**Controllable diffusion models.** Diffusion models have made a huge success in content creation (Ramesh et al., 2022; Song & Ermon, 2019; Ho et al., 2022b; Liu et al., 2022; Ruiz et al., 2023; Nichol et al., 2021; Croitoru et al., 2023; Yang et al., 2022; Wu et al., 2022). ControlNet (Zhang et al., 2023) proposes an architectural design for incorporating spatial conditioning controls into large, pre-trained text-to-image diffusion models using neural networks. GLIGEN (Li et al., 2023) introduces gated attention adapters to take in additional grounding information for image generation. Shape-guided Diffusion (Huk Park et al., 2022) adapts pretrained diffusion models to respond to shape input provided by a user or inferred automatically from the text. Control-A-Video (Chen et al., 2023b) train models to generate videos conditioned on a sequence of control signals, such as edge or depth maps. Despite the impressive progress made by these works, the challenge of generating videos with complex dynamics based solely on text prompts remains unresolved.

**Text-to-video generation.** Although there is a rich literature on video generation (Brooks et al., 2022; Castrejon et al., 2019; Denton & Fergus, 2018; Ge et al., 2022; Hong et al., 2022; Tian et al., 2021; Wu et al., 2021), text-to-video generation remains challenging since it requires the model to synthesize the video dynamics based only on text. Make-A-Video (Singer et al., 2022) breaks down the entire temporal U-Net (Ronneberger et al., 2015) and attention tensors, approximating

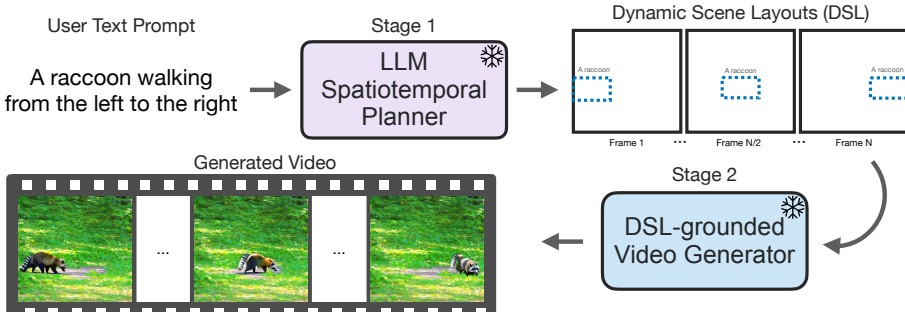

Figure 2: Our method LVD improves text-to-video diffusion models by turning the text-to-video generation into a two-stage pipeline. In stage 1, we introduce an LLM as the spatiotemporal planner that creates plans for video generation in the form of a dynamic scene layout (DSL). A DSL includes objects bounding boxes that are linked across the frames. In stage 2, we condition the video generation on the text and the DSL with a DSL-grounded video generator. Both stages are training-free: *LLMs and diffusion models are used off-the-shelf without updating their parameters.* By using DSL as an intermediate representation for text-to-video generation, LVD generates videos that align much better with the input prompts compared to its vanilla text-to-video model counterpart.

them in both spatial and temporal domains, and establishes a pipeline for producing high-resolution videos. Imagen Video (Ho et al., 2022a) creates high-definition videos through a foundational video generation model and spatial-temporal video super-resolution models. Video LDM (Blattmann et al., 2023) transforms the image generator into a video generator by adding a temporal dimension to the latent space diffusion model. Text2Video-Zero (Khachatryan et al., 2023) introduces two post-processing techniques for ensuring temporal consistency: encoding motion dynamics in latent codes and reprogramming frame-level self-attention with cross-frame attention mechanisms. However, these models still easily fail in generating reasonable video dynamics due to the lack of large-scale paired text-video training data that can cover diverse motion patterns and object attributes, let alone the demanding computational cost to train with such large-scale datasets.

**Grounding and reasoning from large language models.** Several recent text-to-image models propose to feed the input text prompt into an LLM to obtain reasonable spatial bounding boxes and generate high-quality images conditioned on the boxes. LMD (Lian et al., 2023) proposes a training-free approach to guide a diffusion model using an innovative controller to produce images based on LLM-generated layouts. LayoutGPT (Feng et al., 2023) proposes a program-guided method to adopt LLMs for layout-oriented visual planning across various domains. Attention refocusing (Phung et al., 2023) introduces two innovative loss functions to realign attention maps in accordance with a specified layout. Collectively, these methods provide empirical evidence that the bounding boxes generated by the LLMs are both accurate and practical for controlling text-to-image generation. In light of these findings, our goal is to enhance the ability of text-to-video models to generate from prompts that entail complex dynamics. We aim to do this by exploring and harnessing the potential of LLMs in generating spatial and temporal video dynamics. A concurrent work VideoDirectorGPT (Lin et al., 2023) proposes using an LLM in multi-scene video generation, which pursues a different focus and technical route. It explores training diffusion adapters for additional conditioning, while LVD focuses on inference-time guidance and in-depth analysis on the generalization of the LLMs.

## 3 CAN LLMS GENERATE SPATIOTEMPORAL DYNAMICS?

In this section, we explore the extent to which LLMs are able to produce spatiotemporal dynamics that correspond to a specified text prompt. **We aim to resolve three questions in this investigation:**

1. Can LLMs generate realistic dynamic scene layouts (DSLs) aligned with text prompts and discern when to apply specific physical properties?

2. Does the LLM's knowledge of these properties come from its weights, or does the understanding of these concepts develop during inference on the fly?

3. Can LLMs generalize to broader world concepts and relevant properties based on the given examples that only entail limited key properties?

(a) In-context example for **gravity**: A woman walking from the left to the right and a man jumping on the right in a room

(b) In-context example for **elasticity**: A red ball is thrown from the left to the right in a garden

(c) In-context example for **perspective camera projection**: The camera is moving away from a painting

Figure 3: **In-context examples.** We propose to prompt the LLMs with *only one in-context example per key desirable property*. Example **(a)/(b)/(c)** demonstrates gravity/elasticity/perspective projection, respectively. In Section 3, we show LLMs are able to generate DSLs aligned with the query prompts with only these in-context examples, empowering downstream applications such as video generation.

To prompt the LLM for dynamics generation, we query the LLM with a prompt that consists of two parts: task instructions in text and a few examples to illustrate the desired output and rules to follow. Following the prompt, we query the LLM to perform completion, hereby generating the dynamics.

**Task instructions.** We ask the LLM to act as a "video bounding box generator". In the task setup, we outline task requirements like the coordinate system, canvas size, number of frames, and frame speed. We refer readers to our Appendix A.3 for the complete prompt.

**DSL representation.** We ask the LLM to represent the dynamics for each video using a Dynamic Scene Layout (DSL), which is a set of bounding boxes linked across the frames. LLMs are prompted to sequentially generate the boxes visible in each frame. Each box has a representation that includes the location and size of the box in numerical coordinates, as well as a phrase describing the enclosed object. Each box is also assigned a numerical identifier starting from 0, and boxes across frames are matched using their assigned IDs without employing a box tracker.

**In-context examples.** LLMs might not understand our need for real-world dynamics such as gravity in the DSL generation, which can lead to assumptions that may not align with our specific goals. To guide them correctly, we provide examples that highlight the desired physical properties for generating natural spatiotemporal layouts. A key question arises: How many examples do LLMs need in order to generate realistic layouts, given they are not designed for real-world dynamics? One might assume they need numerous prompts to grasp the complex math behind box coordinates and real-world dynamics. Contrary to such intuition, we suggest presenting the LLM with a few examples demonstrating key physical properties and using *only one example for each desired property*. Surprisingly, this often suffices for LLMs to comprehend various aspects including physical properties and camera motions, empowering downstream applications such as video generation (Section 4). We also find that LLMs can extrapolate from provided examples to infer related properties that are not explicitly mentioned. This means we do not need to list every desired property but just highlight a few main ones. Our in-context examples are visualized in Fig. 3, with more details in Appendix A.3.

**Reason before generation.** To improve the interpretability of the LLM output, we ask the LLM to output a brief statement of its reasoning before starting the box generation for each frame. We also include a reasoning statement in each in-context example to match the output format. We refer readers to Appendix A.1 for examples and analysis of reasoning statements from the LLM.

**Investigation setup.** With the potential downstream of video generation, we categorize the desired physical properties into world properties and camera properties. Specifically, we examine gravity and object elasticity as world properties and perspective projection as a camera property. However, the surprising generalization capabilities indicate promising opportunities for LLMs to generalize to other *uninvestigated properties*. As displayed in Fig. 3, we provide the LLM with one example for each of the three properties. We then query the LLM with a few text prompts for both world properties and camera properties (see Fig. 4). We use GPT-4 (OpenAI, 2023) for investigation in this section. Additional models are explored in Section 5, confirming they exhibit similar capabilities.

**Discoveries.** We discovered that for various query scenarios, either seen or unseen, LLMs can create realistic DSLs aligned with the prompt. Notably, this ability is rooted in the LLM's inherent knowledge, rather than solely relying on provided examples. For instance, given the ball example that shows the presence of elasticity in our desired world for LLM to simulate (Fig. 3(b)), LLM understands that the ball, when thrown, will hit the ground and bounce back (Fig. 4(a)), no matter whether it was thrown from the left (Fig. 3(b)) or right. Furthermore, LLM is able to generalize to objects with different elasticity. For example, when replacing the ball with a rock without updating in-context examples (Fig. 4(b)), LLM recognizes that rocks are not elastic, hence they do not bounce, but they still obey gravity and fall. This understanding is reached *without explicit textual cues about the object's nature in either our instructions or our examples*. Since this ability is not derived from the input prompt, it must originate from the LLM's weights. To illustrate our point further, we show

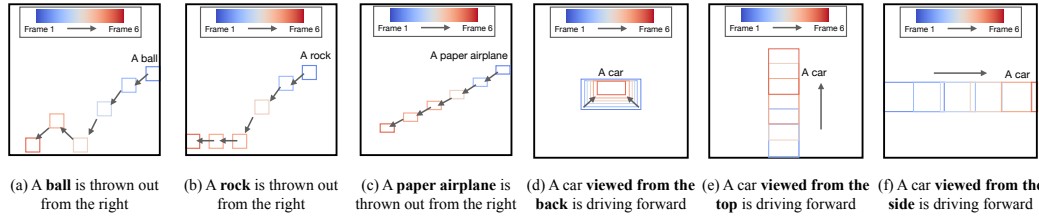

(a) A **ball** is thrown out from the right    (b) A **rock** is thrown out from the right    (c) A **paper airplane** is thrown out from the right    (d) A car **viewed from the back** is driving forward    (e) A car **viewed from the top** is driving forward    (f) A car **viewed from the side** is driving forward

Figure 4: **DSLs generated by the LLM from input text prompts** with only in-context examples in Fig. 3. Remarkably, despite *not having any textual explanations on the applicability of each physical property*, the LLM discerns and applies them appropriately. **(a,b)** The LLM selectively applies the elasticity rule to the ball but not the rock, even though the attributes of a rock are never mentioned in the example. **(c)** The LLM infers world context from the examples and factors in air friction, which is not mentioned either, when plotting the paper airplane's trajectory. **(d-f)** reveal the LLM's inherent grasp of the role of viewpoint in influencing object dynamics, without explicit instructions in the examples. These discoveries suggest that *the LLM's innate knowledge, embedded in its weights, drives this adaptability, rather than solely relying on the provided examples.*

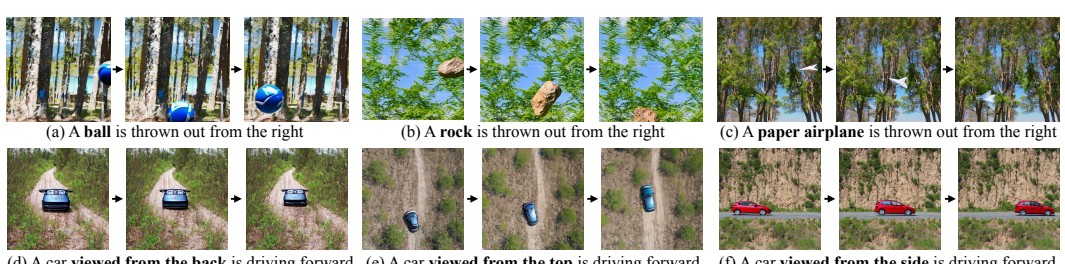

(a) A **ball** is thrown out from the right    (b) A **rock** is thrown out from the right    (c) A **paper airplane** is thrown out from the right

(d) A car **viewed from the back** is driving forward    (e) A car **viewed from the top** is driving forward    (f) A car **viewed from the side** is driving forward

Figure 5: **Videos generated by our method LVD** from the DSLs in Fig. 4. Our approach generates videos that correctly align with input text prompts. **(a-c)** show various objects thrown from the right to the left. **(d-f)** depict objects of the same type viewed from different viewpoints.

the LLM that the "imagined camera" follows the principle of perspective geometry in Fig. 3(c): as the camera moves away from a painting, the painting appears smaller. We then test the LLM with a scenario not directly taught by our examples, as shown in Fig. 4(e,f). Even without explicit explanations about how perspective geometry varies with camera viewpoint and object motion, the LLM can grasp the concept. If the car's forward movement is perpendicular to the direction that the camera is pointing at, perspective geometry does not apply.

More compellingly, by setting the right context of the imagined world, *LLMs can make logical extensions to properties not specified in the instructions or examples*. As shown in Fig. 4(d), the LLM recognizes that the principles of perspective geometry applies not just when a camera moves away from a static object (introduced in Fig. 3(c)), but also when objects move away from a stationary camera. This understanding extends beyond our given examples. As seen in Fig. 4(c), even without explicit mention of air friction, the LLM takes it into account for a paper airplane and generates a DSL in which the paper airplane is shown to fall slower and glide farther than other queried objects. This indicates that our examples help establish an imagined world for the LLM to simulate, removing the need for exhaustive prompts for every possible scenario. LLMs also demonstrate understanding of buoyancy and animals' moving patterns to generate realistic DSLs for Fig. 1 in the reasoning statements. We refer the readers to these statements in Appendix A.1.

These observations indicate that LLMs are able to successfully generate DSLs aligned with complex prompts given only the three in-context examples in Fig. 3. We therefore use these in-context examples and present further quantitative and qualitative analysis in Section 5.

## 4 DSL-GROUNDED VIDEO GENERATION

Leveraging the capability of LLMs to generate DSLs from text prompts, we introduce an algorithm for directing video synthesis based on these DSLs. Our DSL-grounded video generator directs an off-the-shelf text-to-video diffusion model to generate videos consistent with both the text prompt and the given DSL. Our method does not require fine-tuning, which relieves it from the need for instance-annotated images or videos and the potential catastrophic forgetting incurred by training.

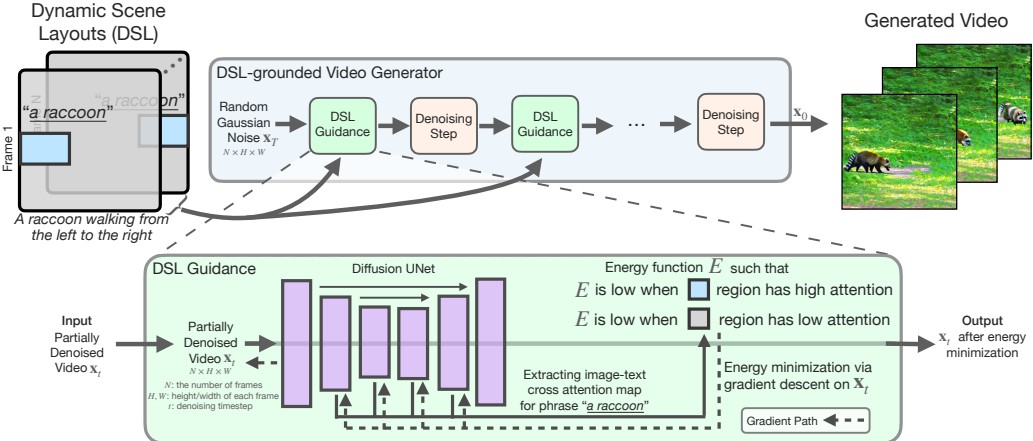

Figure 6: **Our DSL-grounded video generator** generates videos from DSLs using existing text-to-video diffusion models augmented with appropriate DSL guidance. Our method alternates between DSL guidance steps and denoising steps. While the denoising steps gradually transform the random video-shaped Gaussian noise into clean videos, our DSL guidance steps ensure that the denoised videos satisfy our DSL constraints. We designed an energy function minimized when the text-to-video cross-attention maps have high values within the box area and low values outside the box w.r.t. the layout box name. In the DSL guidance step, our method minimizes the energy function $E$ by performing gradient descent on the video, steering the generation to follow the constraints.

A text-to-video diffusion model (*e.g.*, Wang et al. (2023)) generates videos by taking in a random Gaussian noise of shape $N \times H \times W \times C$, where $N$ is the number of frames and $C$ is the number of channels. The backbone network (typically a U-Net (Ronneberger et al., 2015)) iteratively denoises the input based on the output from the previous time step as well as the text prompt, and the text-conditioning is achieved with cross-attention layers that integrate text information into latent features. For each frame, we denote the cross-attention map from a latent layer to the object name in the text prompt by $A \in \mathbb{R}^{H \times W}$, where $H$ and $W$ are the height and width of the latent feature map. For brevity, we ignore the indices for diffusion steps, latent layers, frames, and object indices. The cross-attention map typically highlights positions where objects are generated.

As illustrated in Fig. 6, to make sure an object is generated in and moves with the given DSL, we encourage the cross-attention map to concentrate within the corresponding bounding box. To achieve this, we turn each box of the DSL into a mask $M \in \mathbb{R}^{H \times W}$, where the values are ones inside the bounding box and zeros outside the box. We then define an energy function $E_{\text{topk}} = -\texttt{Topk}(A \cdot M) + \texttt{Topk}(A \cdot (1 - M))$, where $\cdot$ is element-wise multiplication, and $\texttt{Topk}$ takes the average of top-$k$ values in a matrix. Intuitively, minimizing this energy encourages at least $k$ high attention values inside the bounding box, while having only low values outside the box.

However, if the layout boxes in consecutive frames largely overlap (*i.e.*, the object is moving slowly), $E_{\text{topk}}$ may result in imprecise position control. This is because stationary objects in the overlapped region can also minimize the energy function, which the text-to-video diffusion model often prefers to generate. To further ensure that the temporal dynamics of the generated objects are consistent with the DSL guidance, we propose Center-of-Mass (CoM) energy function $E_{\text{CoM}}$ which encourages CoM of $A$ and $M$ to have similar positions and velocities. Denoting the CoM of $t$-th frame's attention map and box mask by $p_t^A$ and $p_t^M$, the $E_{\text{CoM}} = ||p_t^A - p_t^M||_2^2 + ||(p_{t+1}^A - p_t^A) - (p_{t+1}^M - p_t^M)||_2^2$. Overall energy $E$ is a weighted sum of $E_{\text{topk}}$ and $E_{\text{CoM}}$.

To minimize the energy $E$, we add a DSL guidance step before each diffusion step where we calculate the gradient of the energy w.r.t. the partially denoised video for each frame and each object, and update the partially denoised video by gradient descent with classifier guidance (Dhariwal & Nichol, 2021; Xie et al., 2023; Chen et al., 2023a; Lian et al., 2023; Phung et al., 2023; Epstein et al., 2023).

With this simple yet effective training-free conditioning stage, we are able to control video generation with DSL and complete the LVD pipeline of text prompt $\rightarrow$ DSL $\rightarrow$ video. Furthermore, the two stages of our LVD pipeline are implementation-agnostic and can be easily swapped with their counterparts, meaning that the generation quality and text-video alignment will continue to improve with more capable LLMs and layout-conditioned video generators. In Fig. 5, we show that our DSL-grounded video generator can generate realistic videos represented by the DSLs in Fig. 4.

Table 1: **Evaluation of the generated DSLs.** Our LLM spatiotemporal planner is able to generate layouts that align well with the spatiotemporal dynamics requirements in several types of prompts.

| Method | Numeracy | Attribution | Visibility | Dynamics | Sequential | *Average* |
|---|---|---|---|---|---|---|
| Retrieval-based | 20% | 12% | 28% | 8% | 0% | *14%* |
| LVD (GPT-3.5, **ours**) | **100%** | **100%** | **100%** | 71% | 16% | *77%* |
| LVD (GPT-4, **ours**) | **100%** | **100%** | **100%** | **100%** | **88%** | ***98%*** |
| + retrieval examples | 98% | 100% | 100% | 96% | 85% | *96%* |

Table 2: **Detection-based evaluation of the generated videos.** LVD follows the spatiotemporal keywords in the prompts better than ModelScope, the diffusion model that it uses under the hood. Generated videos have a resolution of $256 \times 256$. Additional $512 \times 512$ results are in Appendix A.6.

| Method | LLM-grounded | Numeracy | Attribution | Visibility | Dynamics | Sequential | *Average* |
|---|---|---|---|---|---|---|---|
| ModelScope | ✗ | 32% | 54% | 8% | 21% | 0% | *23.0%* |
| Retrieval-based | ✓ | 15% | 15% | 11% | 9% | 0% | *9.7%* |
| LVD (GPT-3.5, **ours**) | ✓ | **52%** | **79%** | **64%** | 37% | 2% | *46.4%* |
| LVD (GPT-4, **ours**) | ✓ | 41% | 64% | 55% | **51%** | **38%** | ***49.4%*** |

## 5 EVALUATION

**Setup.** We select ModelScope (Wang et al., 2023) as our base video diffusion model and apply LVD on top of it. In addition to comparing LVD with the ModelScope baseline without DSL guidance, we also evaluate LVD against a retrieval-based baseline. Given a text prompt, this baseline performs nearest neighbor retrieval from a large-scale video captioning dataset WebVid-2M based on the similarity between the prompt and the videos with Frozen-In-Time (Bain et al., 2021), and runs GLIP (Li et al., 2022) to extract the bounding boxes for the entities in the captions of the retrieved videos and tracks the bounding boxes with SORT (Bewley et al., 2016) to get the DSLs.

**Benchmarks.** To evaluate the alignment of the generated DSLs and videos with text prompts, we propose a benchmark comprising five tasks: generative numeracy, attribute binding, visibility, spatial dynamics, and sequential actions. Each task contains 100 programmatically generated prompts that can be verified given DSL generations or detected bounding boxes using a rule-based metric, with 500 prompts in total. We generate two videos for each prompt with different random seeds, leading to 1000 video generations per evaluation. We then report the success rate of how many videos follow the text prompts. Furthermore, we adopt the benchmarks (UCF-101 (Soomro et al., 2012) and MSR-VTT (Xu et al., 2016)) as referenced in literature (Blattmann et al., 2023; Singer et al., 2022) to evaluate the quality of video generation. We also conduct an evaluator-based assessment as well. We refer readers to the appendix for details about our evaluation setup and proposed benchmarks.

### 5.1 EVALUATION OF GENERATED DSLS

We evaluate the alignment between the DSLs generated by an LLM and the text prompts without the final video generation. The results are presented in Table 1. We first compare with a baseline model that directly outputs the layout of the closest retrieved video as the DSL without an LLM involved. Its low performance indicates that nearest neighbor retrieval based on text-video similarity often fails to locate a video with matching spatiotemporal layouts. In contrast, our method generates DSLs that align much better with the query prompts, where using GPT-3.5 gives an average accuracy of 77% and using GPT-4 can achieve a remarkable accuracy of 98%. These results highlight that current text-only LLMs can generate reasonable spatiotemporal layouts based solely on a text prompt, which is hard to achieve with a retrieval-based solution. Notably, only three in-context examples in Fig. 3 are used throughout the evaluation. To determine if incorporating examples from real videos enhances the performance, we experimented with adding the samples from retrieval. However, these additional examples lead to similar performance, underscoring the LLM's few-shot generalization capabilities. We also noticed that GPT-3.5 struggles with sequential tasks. While it typically begins correctly, it either skips one of the two actions or executes them simultaneously.

### 5.2 EVALUATION OF GENERATED VIDEOS

**Evaluation of video-text alignment.** We now assess whether the generated videos adhere to the spatial layouts and temporal dynamics described in the text prompts. For evaluation, we also use the rule-based method, with boxes from detector OWL-ViT (Minderer et al., 2022). With results presented in Table 2, ModelScope underperforms in most of the scenarios and is barely able to

Table 3: **Ablations on the energy function used to condition the video diffusion model on the layouts.** We use the same set of layouts generated by the LLM (GPT-4) in the ablation. Neither $E_{\text{ratio}}$ proposed in Chen et al. (2023a) nor $E_{\text{corner}}$ proposed in Boxdiff (Xie et al., 2023) help improve the text-video alignment, whereas simply using $E_{\text{topk}}$ achieves good text-video alignment. Adding $E_{\text{CoM}}$ can further boost the text-video alignment for sequential tasks. Generated videos are of $256 \times 256$.

| $E_{\text{topk}}$ | $E_{\text{ratio}}$ | $E_{\text{corner}}$ | $E_{\text{CoM}}$ | Numeracy | Attribution | Visibility | Dynamics | Sequential | *Average* |
|---|---|---|---|---|---|---|---|---|---|
| ✗ | ✓ | ✗ | ✗ | **44%** | 53% | 11% | **54%** | 21% | *36.4%* |
| ✓ | ✗ | ✓ | ✗ | 38% | 60% | **53%** | 48% | 31% | *45.6%* |
| ✓ | ✗ | ✗ | ✗ | 39% | **65%** | **54%** | 52% | 23% | *46.3%* |
| ✓ | ✗ | ✗ | ✓ | 41% | 64% | **55%** | 51% | **38%** | *49.4%* |

Table 4: **Video quality evaluation.** Lower FVD scores are better. $^\dagger$ indicates our replication of the results. LVD is able to generate videos with higher quality than ModelScope on both benchmarks.

| Method | FVD@UCF-101 ($\downarrow$) | FVD@MSR-VTT ($\downarrow$) |
|---|---|---|
| CogVideo (Chinese) (Hong et al., 2022) | 751 | - |
| CogVideo (English) (Hong et al., 2022) | 702 | 1294 |
| MagicVideo (Zhou et al., 2022) | 699 | 1290 |
| Make-A-Video (Singer et al., 2022) | 367 | - |
| VideoLDM (Blattmann et al., 2023) | 551 | - |
| ModelScope (Wang et al., 2023) | - | 550 |
| ModelScope$^\dagger$ | 984 | 590 |
| ModelScope w/ LVD (**ours**) | 828 | 565 |

generate videos in two of the five tasks. In addition, videos from the retrieved layouts have worse text-video alignment performance when compared to the baseline method. On the other hand, LVD significantly outperforms both baselines by a large margin, suggesting the improved video-text alignment of the proposed LVD pipeline. Interestingly, the layouts generated by GPT-3.5 lead to better text-video alignment for three tasks, as they have less motion in general and are easier to generate videos from. The gap between the results in Table 1 and Table 2 still indicates that the bottleneck is in the DSL-grounded video generation stage.

**Ablations on the energy function.** $E$. Boxdiff (Xie et al., 2023) proposes a corner term $E_{\text{corner}}$ with $E_{\text{topk}}$, and Chen et al. (2023a) proposes to use a ratio-based energy function $E_{\text{ratio}}$. As shown in Table 3, we found that using only $E_{\text{topk}}$ achieves a good text-video alignment already. We also recommend adding $E_{\text{CoM}}$ when sequential tasks are entailed, as it offers significant boosts.

**Evaluation of video quality.** Since our primary objective is to generate videos that align with intricate text prompts without additional training, improving video quality is not the key objective to demonstrate the effectiveness of our method. We evaluate the video quality through FVD score on UCF-101 and MSR-VTT. The FVD score measures if the generated videos have the same distribution as the dataset. Both benchmarks feature videos of complex human actions or real-world motion. Since our method is agnostic to the underlying models and applicable to different text-to-video diffusion models, we mostly compare with ModelScope (Wang et al., 2023), the diffusion model that we use under the hood. Both LVD and our replicated ModelScope generate $256 \times 256$ videos. As shown in Table 4, LVD improves the generation quality compared to the ModelScope baseline on both datasets, suggesting our method can produce actions and motions that are visually more similar to real-world videos. ModelScope and LVD surpass models such as Zhou et al. (2022) on MSR-VTT, while being worse on UCF-101, possibly due to different training data.

**Visualizations.** In Fig. 7, we show our video generation results along with baseline Modelscope. LVD consistently outperforms the base model in terms of generating the correct number of objects, managing the introduction of objects in the middle of the video, and following the sequential directions specified in the prompt. In addition, LVD is also able to handle descriptions of multiple objects well. For example, in Fig. 7(b), while ModelScope assigns the color word for the dog to a ball, LVD precisely captures both the blue dog and the brown ball in the generation.

**Evaluator-based assessment.** To further validate the quality of videos generated with LVD, we conducted an evaluator-based assessment. We asked two questions to 10 participants about each of the 20 pairs of videos randomly selected from our benchmark: "*Which video aligns better with the text prompt?*" and "*Which video has better quality?*". Each set is composed of two videos, one generated by baseline and the other by LVD. We randomly shuffled the videos within each set. For

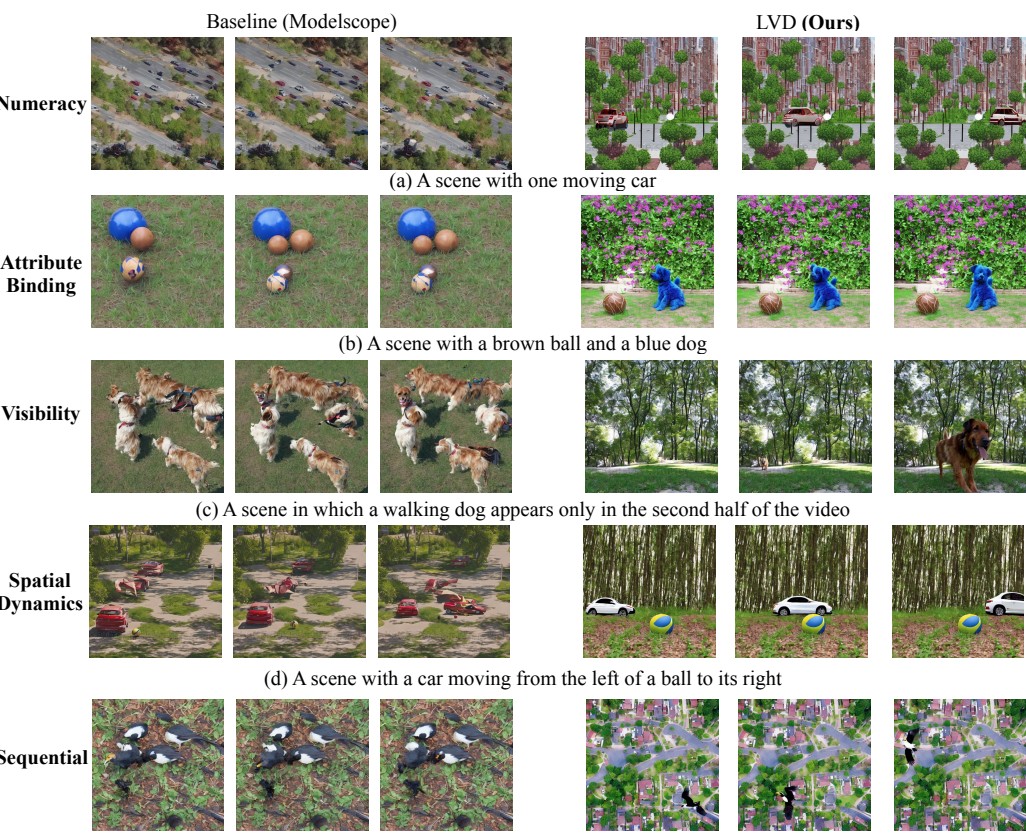

Baseline (Modelscope)                                           LVD **(Ours)**

**Numeracy**

(a) A scene with one moving car

**Attribute Binding**

(b) A scene with a brown ball and a blue dog

**Visibility**

(c) A scene in which a walking dog appears only in the second half of the video

**Spatial Dynamics**

(d) A scene with a car moving from the left of a ball to its right

**Sequential**

(e) A top-down viewed scene in which a bird initially on the lower right of the scene. It first moves to the lower left of the scene and then moves to the upper left of the scene.

Figure 7: Despite not aiming for each individual aspect of the quality improvement, **LVD generates videos that align much better with the input prompts** in several tasks that require numerical, temporal, and 3D spatial reasoning. Best viewed when zoomed in.

Table 5: **Assessment based on human evaluators.** Despite sharing *the same diffusion weights* as the baseline model (Wang et al., 2023), our method is preferred significantly more often.

| Question | Baseline | LVD | Similar |
|---|---|---|---|
| *Which video aligns better with the text prompt?* | 3.2% | **85.8**% | 12.6% |
| *Which video has better quality?* | 14.5% | **66.0**% | 19.5% |

each question, the options are "Video #1", "Video #2", and "Similar". We show the averaged score in Table 5. LVD holds a substantial edge over the baselines. In 96.8% of alignment cases and 85.5% of visual quality cases, the evaluators indicate that LVD outperforms or is at least on par with the baseline model. This highlights that LVD brings about a significant enhancement in the performance of baseline video diffusion models in both the alignment and visual quality.

## 6    DISCUSSION AND FUTURE WORK

In this work, we propose LLM-grounded Video Diffusion (LVD) to enhance the capabilities of text-to-video diffusion models to handle complex prompts without any LLM or diffusion model parameter updates. Although LVD shows substantial enhancement over prior methods, there is still potential for further improvement. For example, as shown in the Appendix A.5, LVD still inherits limitations from the base model when it comes to synthesizing objects or artistic styles it struggles with. We anticipate that techniques like LoRA fine-tuning (Hu et al., 2021) could help mitigate such challenges and leave this exploration for future research. Our aim for this work is **1)** to demonstrate LLM's surprising spatiotemporal modeling capability and **2)** to show that LLM-grounded representation, such as DSLs, can be leveraged to improve the alignment for text-to-video generation, with potential extensions to control finer-grained properties, including human poses, to model complex dynamics. Our DSL-to-video stage is still simple, and thus a more advanced DSL-conditioned video generator may further improve the precision of control and video quality in future work.

## REPRODUCIBILITY STATEMENT

We mainly apply and evaluate our method on open-sourced model ModelScope (Wang et al., 2023), allowing the reproduction of results in this work. We present our prompts for DSL generation in Appendix A.3. We present implementation and benchmarking details in Appendix A.4. We will also release code and benchmarks for reproducibility and future research.

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

## A APPENDIX

### A.1 REASONING STATEMENTS FROM GPT-4

As described in Section 3, we ask the model to output a reasoning statement for interpretability. We present examples of the reasoning statements for DSL generation in Table 6. The corresponding DSLs and generated videos are presented in Fig. 4, Fig. 1, and Fig. 5. The first three statements are from ChatGPT web interface. The last four statements are from OpenAI API. All statements are from GPT-4.

These statements provide useful insights about why GPT-4 generates these DSLs in specific ways. For example, the first reasoning statement indicates that GPT-4 takes the flying pattern of birds into account. The second statement shows that GPT-4 applys the buoyancy for the wooden barrel. GPT-4 also considers the reaction of the cat in the third statement. For the fourth statement, GPT-4 demonstrates its understanding that the rock, unlike the ball provided in the in-context example, does not bounce (note that our y-coordinate is top-to-bottom as introduced in the text instructions, so it is correct that the y-coordinate increases).

Interestingly, we observe that GPT-4 with ChatGPT web interface often outputs longer reasoning statements compared to GPT-4 called with APIs. We hypothesize that GPT-4 used in ChatGPT interface has prompts encouraging more explanations rather than following the patterns in the examples.

### A.2 ADDITIONAL QUALITATIVE VISUALIZATIONS

We present additional qualitative visualizations of LVD in Fig. 8.

**Handling ambiguous prompts.** As shown in Fig. 8(a), our LVD can successfully generate videos for "a walking dog", which is a prompt that has ambiguity, as the walking direction of the dog is not given. Despite the missing information, the LLM is still able to make a reasonable guess for the walking direction in DSL generation, leading to reasonable video generation. Furthermore, this assumption is included in the reasoning statement:

```
Reasoning:  A dog walking would have a smooth, continuous motion
across the frame.  Assuming it is walking from left to right, its
x-coordinate should gradually increase.  The dog's y-coordinate
should remain relatively consistent, with minor fluctuations to
represent the natural motion of walking.  The size of the bounding
box should remain constant since the dog's size doesn't change.
```

This explicitly indicates that the LLM can detect the missing information and make reasonable assumptions for generation.

**Handling overlapping boxes.** Our DSL-to-video stage can process highly overlapping boxes. In addition to video frames, we also show the dynamic scene layouts generated by the LLM in dashed boxes (white for the raccoon and blue for the barrel) in Fig. 8(b). Although the box of the raccoon overlaps about half of the barrel, the generation of the raccoon on the barrel is still realistic and the motion still makes sense.

**Handling dynamic background.** ModelScope often prefers to generate a background with little motion. Although this preference is inherited by our training-free method, our model still works when asked to generate a background that changes significantly. For example, in Fig. 8(c) we query the model to generate a brown bear taking a shower in a lake when the lightning strikes, and our method generates both successfully, with significant variations on the background. Note that the model generates the effect of rain when only lightning is mentioned in the prompt, which is a reasonable assumption inherited from our base model.

### A.3 OUR PROMPTS AND IN-CONTEXT EXAMPLES FOR DSL GENERATION

Table 10 presents the prompts that we use for DSL generation in the quantitative evaluation. To generate DSLs, we replace the "User Text Prompt for DSL Generation" in the template with the text query that we would like to generate DSLs from and pass the prompt to an LLM. We use

Table 6: Reasoning statements from GPT-4.

| | Query Prompt | LLM Reasoning statements |
|---|---|---|
| 1 | A bird is flying from the left to the right | A bird flying from left to right will have its x-coordinate steadily increasing. Birds tend to fly in a somewhat leveled manner, but they can have slight vertical movements (up and down) due to flapping or gliding. Given the short timeframe of six frames, we'll assume the bird has a slight upward movement initially and then a glide downward, reflecting a flap-glide pattern typical for many birds. |
| 2 | A wooden barrel is drifting on a river | A wooden barrel drifting on a river will follow the flow direction, typically moving from left to right or right to left. The barrel, being buoyant, will remain on the surface and will be subjected to the river's currents. As such, the x-coordinate of the barrel will change, indicating its drift, but its y-coordinate will remain largely consistent since it is floating on the surface. For simplicity, let's assume the river flows from left to right. |
| 3 | A raccoon is walking towards a cat | The raccoon starts from one side and moves closer to the cat. As both the raccoon and the cat are made larger, their bounding boxes will be more extensive. The raccoon's x-coordinate should increase and it should get closer to the x-coordinate of the cat. The cat might be stationary at first, but as the raccoon approaches, it might move backward a bit due to its natural instinct to maintain a distance. Both the raccoon and the cat will have a constant y-coordinate because they are on the same ground level. |
| 4 | A rock is thrown out from the right | A rock is thrown from the right, so its x-coordinate should decrease. Due to gravity, its y-coordinate should increase, and the speed should be faster in later frames until it hits the ground. Unlike a ball, a rock does not bounce back when it hits the ground. |
| 5 | A car viewed from the side is driving forward | As the car is driving forward, its x-coordinate should increase over time. The y-coordinate should remain constant as the car is moving on a flat surface. The size of the car should remain constant as it is viewed from the side. |
| 6 | A car viewed from the top is driving forward | The car is moving forward, so its y-coordinate should decrease. The x-coordinate should remain the same as there is no lateral movement. The car's size remains the same as the camera is stationary. |
| 7 | A car viewed from its back is driving forward | As the car is driving forward and the camera is viewing from the back, the car will appear to get smaller as it moves away from the camera. The x-coordinate and y-coordinate will remain the same, but the width and height of the bounding box will decrease to represent the car moving away. |

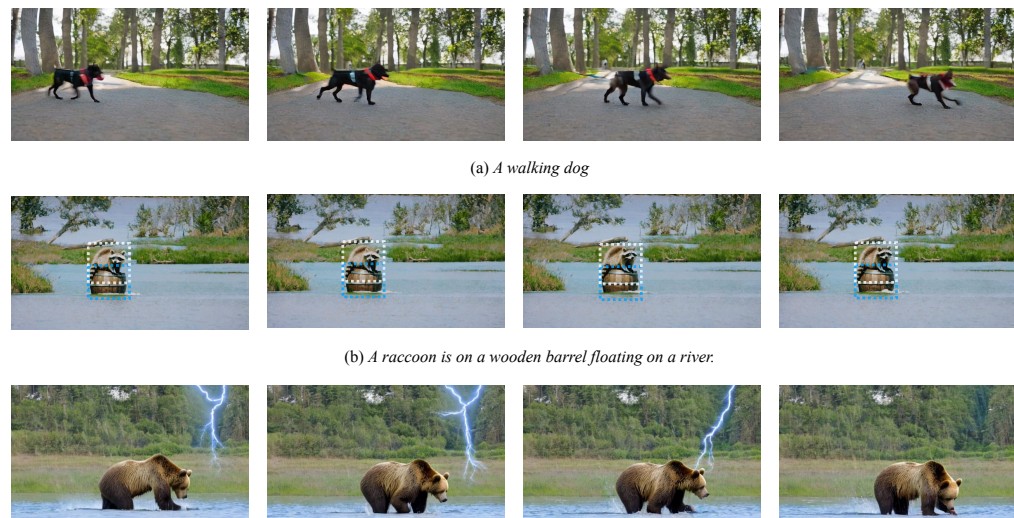

(a) *A walking dog*

(b) *A raccoon is on a wooden barrel floating on a river.*

(c) *A brown bear is taking a shower in a lake, lightning strikes*

Figure 8: Additional visualizations of LVD. **(a)** Given a prompt that has ambiguity (the walking direction of the dog is not given), LLM is still able to make a reasonable guess and often includes the assumption in the reasoning statement, leading to reasonable generation. **(b)** Our DSL-to-video stage is able to process highly overlapping boxes (**outlined in the dashed boxes**): the box of the raccoon overlaps about half of the barrel, yet the generation of the raccoon on the barrel is still realistic and not abrupt. **(c)** Although our base video diffusion model ModelScope often generates static background, which is inherited by our training-free method, our model still works when asked to generate background that changes significantly such as lightning and raining. Note that the model generates raining when only lightning is mentioned in the prompt, which is a reasonable assumption.

the chat completion API[1] for both GPT-3.5 and GPT-4. For API calls, line 1-3 are assigned with role "system". Each caption line in the in-context examples and user query are assigned with role "user". The example outputs are assigned with role "assistant". The final "Reasoning" line is omitted. For querying through ChatGPT web interface [2], we do not distinguish the roles of different lines. Instead, we merge all the lines into one message. The benchmarks are conducted by querying the LLMs through the API calls. The visualizations in Fig. 1 are conducted by querying the GPT-4 through the ChatGPT web interface (with horizontal videos generated by Zeroscope[3], a model fine-tuned from ModelScope optimized for horizontal video generation).

## A.4 DETAILS ABOUT OUR SETUP AND BENCHMARKS

**Setup.** Since our method is training-free, we introduce our inference setup in this section. For DSL-grounded video generation, we use DPMSolverMultiStep scheduler (Lu et al., 2022a;b) to denoise 40 steps for each generation. We use the same hyperparams as the baselines, except that we employ DSL guidance. For ModelScope (Wang et al., 2023), we generate videos of 16 frames. For qualitative assessment in our benchmarks, we generate videos with resolution of both $256 \times 256$ and $512 \times 512$ for comprehensive comparisons. For FVD evaluation, we generate videos with resolution $256 \times 256$, following our baseline. For evaluator-based assessment, we generate videos with resolution $512 \times 512$ with only $E_{\text{topk}}$. For Zeroscope examples in visualizations, we generate $576 \times 320$ videos of 24 frames. For DSL guidance, we scale our energy function by a factor of $5$. We perform DSL guidance 5 times per step only in the first 10 steps to allow the model to freely adjust the details generated in the later steps. We apply a background weight of $4.0$ and a foreground weight of $1.0$ to each of the terms in the energy function, respectively. The $k$ in Topk was selected by counting $75\%$ of the positions in the foreground/background in the corresponding term, inspired by

---

[1] https://platform.openai.com/docs/guides/gpt/chat-completions-api
[2] https://chat.openai.com/
[3] https://huggingface.co/cerspense/zeroscope_v2_576w

previous work for image generation (Xie et al., 2023). $E_{\text{CoM}}$ is weighted by 0.03 and added to $E_{\text{topk}}$ to form the final energy function $E$. The energy terms for each object, frame, and cross-attention layers are averaged. The "learning rate" for the gradient descent follows $\sqrt{1 - \hat{\alpha}_t}$ for each denoising step $t$, where the notations are introduced in Dhariwal & Nichol (2021).

**Proposed benchmark.** To evaluate the alignment of the generated DSLs and videos with text prompts, we propose a benchmark comprising five tasks: generative numeracy, attribute binding, visibility, spatial dynamics, and sequential actions. Each task contains 100 programmatically generated prompts that can be verified given DSL generations or detected bounding boxes using a rule-based metric, with 500 text prompts in total. We then generate 2 videos for each text prompt, resulting in 1000 videos generated. For LLM-grounded experiments, we generate one layout per prompt and generate two videos per layout.

The generative numeracy prompts are generated from template `A realistic lively video of a scene with [number] [object type]`, where object types are sampled from `['car', 'cat', 'bird', 'ball', 'dog']`.

The attribute binding prompts are generated from template `A realistic lively video of a scene with a [color1] [object type1] and a [color2] [object type2]` where colors are sampled from `['red', 'orange', 'yellow', 'green', 'blue', 'purple', 'pink', 'brown', 'black', 'white', 'gray']`.

The visibility prompts are generated from template `A realistic lively video of a scene in which a [object type] appears only in the [first/second] half of the video`.

The spatial dynamics prompts are generated from template `A realistic lively video of a scene with a [object type] moving from the [left/right] to the [right/left]` and template `A realistic lively video of a scene with a [object type1] moving from the [left/right] of a [object type2] to its [right/left]`.

The sequential actions prompts are generated from template `A realistic lively video of a scene in which a [object type1] initially on the [location1] of the scene. It first moves to the [location2] of the scene and then moves to the [location3] of the scene`. The three locations are sampled from `[('lower left', 'lower right', 'upper right'), ('lower left', 'upper left', 'upper right'), ('lower right', 'lower left', 'upper left'), ('lower right', 'upper right', 'upper left')]`.

**UCF-101 and MSR-VTT benchmarks.** For UCF-101, we generate 10k videos, each of 16 frames and $256 \times 256$ resolution, with either the baseline or our method and using captions randomly sampled from UCF-101 data, and evaluate FVD between the 10k generated videos and videos in UCF-101 following (Blattmann et al., 2023; Singer et al., 2022). We first generate a template prompt for each class, use LLM to generate multiple sets of layouts for each prompt so that there are in total 10k video layouts from all the prompts following the class distribution of the training set. We then calculate the FVD score with the training set of UCF-101. The video resolution is $256 \times 256$ and each video contains 16 frames. For evaluation on MSR-VTT, we use LLM to generate layout for each caption of each video in the test set, which contains 2990 videos. We then randomly select 2048 layouts, generate videos for each layout, and calculate the FVD score with the test set videos.

### A.5 FAILURE CASES

As shown in Fig. 10 (left), the base model (Wang et al., 2023) does not generate high-quality cartoons for "an apple moving from the left to the right", with the apples in weird shapes as well as random textures in the background. Although our model is able guide the motion of the apple generation with DSL-guidance, shown in Fig. 10 (right), our model still inherits the other problems that are present in the baseline generation. Techniques that can be used for enhancing the ability for domain-specific generation such as performing LoRA fine-tuning (Hu et al., 2021) on the diffusion models may alleviate such issues. Since the scope of this work does not include fine-tuning diffusion models for higher quality generation, we leave the investigation to future work.

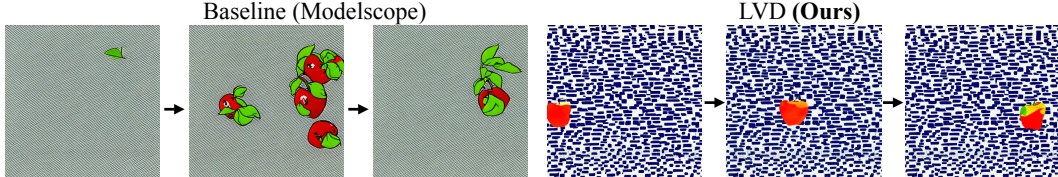

Figure 10: **One failure case**: *A cartoon of an apple moving from the left to the right.* Since LVD is training-free, it does not improve the quality on domains/styles that the base model is not good at. Although LVD correctly generates a video that is aligned with the prompt, neither the baseline or LVD generates background that fits the apple in the foreground.

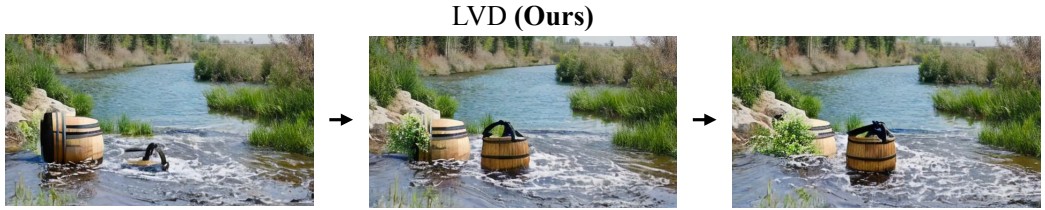

Figure 11: **Another failure case**: *A wooden barrel is drifting on a river.* Since LVD controls the cross-attention map between the image and the text, the DSL guidance sometimes leads to undesirable generations during the energy minimization process. In this failure case, the model attempts to hide the barrel on the left with leaves and pop a barrel up from the water to reach the desired energy minimization effect, while the DSL means to move the barrel to the right.

Another failure case is shown in Fig. 11, in which the DSL grounding process actually minimizes the energy function in an unexpected way. Instead of moving the barrel to the right in order to render the drifting effect, the model attempts to hide the barrel on the left with leaves and pop a barrel up from the water to reach the desired energy minimization effect in the DSL grounding process. Reducing the ambiguity caused by cross-attention-based control is needed to resolve this problem. We leave its investigation as a future direction.

We also show a failure case of LLM generating DSLs. From Table 1 we show taht DSLs generated by GPT4 have 100% accuracy in all categories except for sequential movements. This means LLM can still fail when generating layouts for long-horizon dynamics. Here we show a failure case in Figure 9. The prompt is *A ball first moves from lower left to upper left of the scene, then moves to upper right of the scene.* We can see that instead of moving to top left, the generated box only moves to slightly above the middle of the frame box, which is inconsistent with the prompt. Since LLM queries are independent, we found that this accounts for most of the fluctuations in the performance in our layout generation stage.

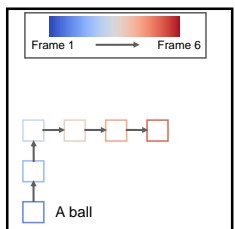

Figure 9: Failure case of LLM-generated DSL.

A.6 ADDITIONAL QUALITATIVE RESULTS WITH HIGHER VIDEO RESOLUTION

We also present additional results on our proposed benchmark in Table 7 with resolution $512 \times 512$. We found that as the resolution increases, the performance gaps between our method and baselines are more significant, which is likely because it is easier to control the motion with high-resolution latents and the detection boxes are more accurate and have lower levels of noise in evaluation. Our full method, with $E_{CoM}$, surpasses the baseline without guidance by a large margin. We also ablated other choices of energy functions, and show that $E_{ratio}$, proposed in Chen et al. (2023a), leads to degradation in performance. Adding $E_{corner}$ proposed in Xie et al. (2023) leads to similar performance in terms of text-video alignment. Since different types of energy terms are involved, we also experimented with giving more weights to $E_{corner}$. However, the performance fluctuates between $59.2\%$ and $60.9\%$, as we give $1\times$ to $10\times$ more weights to $E_{corner}$. We hypothesize that the spatial normalization term in $E_{corner}$ leads to its insignificant change in performance, and thus we also implemented a version

Table 7: **Additional results and ablations on videos of higher resolution.** Generated videos are of $512 \times 512$. † indicates unnormed corner energy function.

| | $E_{\text{topk}}$ | $E_{\text{ratio}}$ | $E_{\text{corner}}$ | $E_{\text{CoM}}$ | Numeracy | Attribution | Visibility | Dynamics | Sequential | *Average* |
|---|---|---|---|---|---|---|---|---|---|---|
| No guidance | ✗ | ✗ | ✗ | ✗ | 8% | 66% | 2% | 19% | 0% | *18.6%* |
| With only $E_{\text{ratio}}$ | ✗ | ✓ | ✗ | ✗ | 29% | 75% | 13% | 49% | 23% | *37.5%* |
| With $E_{\text{corner}}$ | ✓ | ✗ | ✓ | ✗ | **64%** | **94%** | 47% | 66% | 36% | *60.9%* |
| With $E_{\text{corner}}^{\dagger}$ | ✓ | ✗ | ✓ | ✗ | 17% | 86% | 10% | 36% | 5% | *30.6%* |
| With only $E_{\text{topk}}$ | ✓ | ✗ | ✗ | ✗ | **62%** | **92%** | 48% | 65% | 32% | *59.6%* |
| With $E_{\text{CoM}}$ | ✓ | ✗ | ✗ | ✓ | 56% | **92%** | **62%** | **75%** | **52%** | ***67.2%*** |

Table 8: **Ablations on the number of in-context examples.** We found that 3 in-context examples are sufficient for stable performance. †: We observe instabilities when we only use one example with GPT-3.5, where two of the three runs could not finish due to invalid DSLs generated.

| Number of in-context examples | 1 | 3 (default) | 5 |
|---|---|---|---|
| DSL evaluation accuracy (GPT-4) | 96.2%±1.9%$^{\dagger}$ | 97.3%±0.6% | 96.6% |

Table 9: **Ablations on the number of DSL guidance steps between two denoising steps in LVD.** We use 5 DSL guidance steps as the performance saturates with more DSL guidance steps.

| DSL guidance steps | 1 | 3 | 5 (default) | 7 |
|---|---|---|---|---|
| Detection-based evaluation accuracy | 35.3% | 46.9% | 49.4% | 49.4% |

of the $E_{\text{corner}}$ that removed the spatial normalization. However, we observe that this significantly degrades the text-video alignment compared to using only $E_{\text{topk}}$.

## A.7 ADDITIONAL ABLATION EXPERIMENTS

### A.7.1 ABLATIONS ON THE LLM-GROUNDED DSL GENERATION STAGE

**In-context examples.** We test prompts with 1, 3, and 5 in-context examples. We first test the performance of the LLM-grounded DSL generator with each of the 3 in-context examples that we composed. As shown in Table 8, we find that GPT-4 could still generate high-quality DSLs with only 1 in-context example, despite slightly degraded performance and increased performance fluctuation. However, for GPT-3.5, we observe instabilities that two of three runs could not finish the whole benchmark within three attempts for each prompt, mainly because they are not able to generate DSLs that can be parsed (*e.g.*, some DSLs do not have boxes for all the frames). Therefore, we still recommend at least three examples. Furthermore, we composed two additional in-context examples and evaluated the quality of the DSLs with **1)** 3 randomly chosen DSLs from the 5 DSLs, and **2)** all 5 DSLs as in-context examples. We find that our method is stable with different sets of in-context examples when 3 examples are given. In addition, we did not find additional benefits from adding more DSLs. We therefore keep 3 DSLs as our default choice.

### A.7.2 ABLATIONS ON THE DSL-GROUNDED VIDEO GENERATION STAGE

**DSL guidance steps.** We test 1, 3, 5, and 7 steps of DSL guidance between denoising steps, respectively. The results are shown in Table 9. We can see that the average accuracy improves when adding more DSL guidance steps from 1 to 5 steps, and the performance plateaus when we use more than 5 steps. This means 5 steps is enough and probably the best choice considering the trade-off between performance and efficiency. We therefore use 5 DSL guidance steps.

Table 10: Our prompt for DSL generation. When using API calls, line 1-3 presented to GPT-4 with role "system". Each caption line in the in-context examples and user query are assigned with role "user". The example outputs are assigned with role "assistant". These lines are merged into one message when querying GPT-4 through the ChatGPT web interface.

```
1 You are an intelligent bounding box generator for videos. You don't need
     to generate the videos themselves but need to generate the bounding
     boxes. I will provide you with a caption for a video with six
     frames, with two frames per second. Your task is to generate a list
     of realistic bounding boxes for the objects mentioned in the caption
     for each frame as well as a background keyword. The video frames are
     of size 512x512. The top-left corner has coordinates [0, 0]. The
     bottom-right corner has coordinates [512, 512]. The bounding boxes
     should not overlap or go beyond the frame boundaries.
2
3 Each frame should be represented as '[{'id': unique object identifier
     incrementing from 0, 'name': object name, 'box': [box top-left
     x-coordinate, box top-left y-coordinate, box width, box height]},
     ...]'. Each box should not include more than one object. Your
     generated frames must encapsulate the whole scenario depicted by the
     caption. Assume objects move and interact based on real-world
     physics, considering aspects such as gravity and elasticity. Assume
     the camera follows perspective geometry. Boxes for an object should
     have the same id across the frames, even if the object may disappear
     and reappear. If needed, you can make reasonable guesses. Provide a
     concise reasoning statement that is not longer than a few sentences
     before each generation. Refer to the examples below for the desired
     format. Never use markdown or other formats not in the examples. Do
     not start each frame with '-'. Do not include any comments in your
     response.
4
5 [In-context examples]
6
7 Caption: {User Text Prompt for DSL Generation}
8 Reasoning:
```

Table 11: Our in-context examples for DSL generation. We use these three in-context examples that we use throughout our work unless stated otherwise.

```
1 Caption: A woman walking from the left to the right and a man jumping on
      the right in a room
2 Reasoning: A woman is walking from the left to the right so her
      x-coordinate should increase with her y-coordinate fixed. A man is
      jumping on the right so his x-coordinate should be large, and his
      y-coordinate should first decrease (upward movement) and then
      increase (downward movement due to gravity).
3 Frame 1: [{'id': 0, 'name': 'walking woman', 'box': [0, 270, 120, 200]},
      {'id': 1, 'name': 'jumping man', 'box': [380, 290, 120, 180]}]
4 Frame 2: [{'id': 0, 'name': 'walking woman', 'box': [50, 270, 120,
      200]}, {'id': 1, 'name': 'jumping man', 'box': [380, 205, 120, 200]}]
5 Frame 3: [{'id': 0, 'name': 'walking woman', 'box': [100, 270, 120,
      200]}, {'id': 1, 'name': 'jumping man', 'box': [380, 175, 120, 200]}]
6 Frame 4: [{'id': 0, 'name': 'walking woman', 'box': [150, 270, 120,
      200]}, {'id': 1, 'name': 'jumping man', 'box': [380, 175, 120, 200]}]
7 Frame 5: [{'id': 0, 'name': 'walking woman', 'box': [200, 270, 120,
      200]}, {'id': 1, 'name': 'jumping man', 'box': [380, 205, 120, 200]}]
8 Frame 6: [{'id': 0, 'name': 'walking woman', 'box': [250, 270, 120,
      200]}, {'id': 1, 'name': 'jumping man', 'box': [380, 290, 120, 180]}]
9 Background keyword: room
10
11 Caption: A red ball is thrown from the left to the right in a garden
12 Reasoning: A ball is thrown from the left to the right, so its
      x-coordinate should increase. Due to gravity, its y-coordinate
      should increase, and the speed should be faster in later frames
      until it hits the ground. Due to its elasticity, the ball bounces
      back when it hits the ground.
13 Frame 1: [{'id': 0, 'name': 'red ball', 'box': [0, 206, 50, 50]}]
14 Frame 2: [{'id': 0, 'name': 'red ball', 'box': [80, 246, 50, 50]}]
15 Frame 3: [{'id': 0, 'name': 'red ball', 'box': [160, 326, 50, 50]}]
16 Frame 4: [{'id': 0, 'name': 'red ball', 'box': [240, 446, 50, 50]}]
17 Frame 5: [{'id': 0, 'name': 'red ball', 'box': [320, 366, 50, 50]}]
18 Frame 6: [{'id': 0, 'name': 'red ball', 'box': [400, 446, 50, 50]}]
19 Background keyword: garden
20
21 Caption: The camera is moving away from a painting
22 Reasoning: Due to perspective geometry, the painting will be smaller in
      later timesteps as the distance between the camera and the object is
      larger.
23 Frame 1: [{'id': 0, 'name': 'painting', 'box': [156, 181, 200, 150]}]
24 Frame 2: [{'id': 0, 'name': 'painting', 'box': [166, 189, 180, 135]}]
25 Frame 3: [{'id': 0, 'name': 'painting', 'box': [176, 196, 160, 120]}]
26 Frame 4: [{'id': 0, 'name': 'painting', 'box': [186, 204, 140, 105]}]
27 Frame 5: [{'id': 0, 'name': 'painting', 'box': [196, 211, 120, 90]}]
28 Frame 6: [{'id': 0, 'name': 'painting', 'box': [206, 219, 100, 75]}]
29 Background keyword: room
```

