# OpenReview forum: "LLM-grounded Video Diffusion Models"
_ICLR.cc/2024/Conference — ICLR 2024 poster_

### Official Review · Reviewer_ZTWf · 2023-11-01

**Soundness:** 3 good
**Presentation:** 4 excellent
**Contribution:** 3 good
**Rating:** 6
**Confidence:** 3

**Summary:**

In this paper, the authors propose a new text-to-video generation pipeline called LLM-grounded Video Diffusion (LVD). In particular, it first uses LLM to generate the layouts of the video and then uses the generated layout to guide a pre-trained video diffusion model. The whole process does not update the weights of both the LLM and video diffusion model. Besides, the authors show that LLMs’ can generate spatiotemporal dynamics aligned with text prompts in a few-shot setting. Qualitative results and quantitative results show that LVD generates higher quality videos that also align more with text.

**Strengths:**

- Overall, the paper is well-organized and easy to follow. The figures and tables are informative.

- The finding that LLMs can generate good spatiotemporal dynamics with only three examples is interesting and well supported by the experiments. The exploration of physical properties contained in LLM is also inspiring and deserves further research.

- The results generated by LVD are promising compared to the baseline, ModelScope.

**Weaknesses:**

- The idea of using LLM to generate layout is already explored in LayoutGPT (Feng et al., 2023) and LMD (Lian et al., 2023). LMD also adapts in a training-free manner. It is beneficial for the authors to include a more detailed comparison.

- The technical contribution is limited. The first layout generation part is similar to LMD, and the second part is a straightforward application of other training-free layout-to-image methods in the video domain.

**Questions:**

- From Table 3, we can see that LVD improves the video quality. What causes the improvement?

- Is LLM able to generate reliable layouts using text without direction information, such as “a walking dog”.

---

> ### Author Response · Authors · 2023-11-19
>
> Thank you for your encouraging review! Here we address your comments individually:
>
> > The idea of using LLM to generate layout is already explored in LayoutGPT (Feng et al., 2023) and LMD (Lian et al., 2023).
>
> > The first layout generation part is similar to LMD, and the second part is a straightforward application of other training-free layout-to-image methods in the video domain.
>
> Thanks for the comment. However, we would like to point out a few differences in terms of findings and the proposed methods between our work and LMD.
>
> As for the LLM-grounded dynamic scene layout generation stage, **there are two noteworthy distinctions between our work and previous works, as explained in detail in the General Rebuttal \[Q2\]**.
> 1. Most of text-to-image-layout generation tasks introduced in previous works (LayoutGPT and LMD) can be completed without in-depth understanding of the properties of the objects in the prompt. In contrast, even a simple generation of dynamic scene layouts of "a falling object" requires understanding several world properties, object properties, and camera properties, which is **highly non-trivial for an LLM and not tested by previous works**.
> 2. More importantly, previous works either only evaluate properties that are **already seen in in-context examples** (e.g., the 4 main tasks that LMD showcases are already demonstrated in the in-context examples in Table 10 of the LMD paper), or **retrieve related layouts from a layout database** as in-context examples (LayoutGPT). **Neither work demonstrates whether LLMs have the knowledge for generalization without relevant in-context examples.** In contrast, in Section 3 (Discoveries), we show that LLMs can generalize to physical properties or object properties that are **unseen in the in-context examples** (e.g., inferring the property of a paper airplane, while similar concepts are never mentioned in the in-context examples). This implies LLMs have an intrinsic understanding of the physical world instead of just emulating from in-context examples.
>
> As for the dynamic scene layout to video generation stage, **as explained in the General Rebuttal \[Q1\],** we propose a temporal energy term that provides guidance on the positions and velocities of center-of-mass of each object, which significantly improves our generation accuracy, especially on the temporal tasks. **Since previous layout-to-image methods only focus on generating one image (i.e., do not take temporal consistency into accounts), we believe our introduction of temporal center-of-mass energy term sets our method further apart from previous image layout conditioning methods that you mentioned.** We have updated our paper PDF to include the descriptions to this energy term.
>
> **We hope the reviewer can agree with us on these significant distinctions that are either not applicable to previous LLM-grounded text-to-image works or are not demonstrated by them.** We will update the manuscript accordingly to express these distinctions more clearly.

---

> ### Author Response · Authors · 2023-11-19
>
> > From Table 3, we can see that LVD improves the video quality. What causes the improvement?
>
> Our method indeed improves on the FVD scores on both UCF-101 and MSR-VTT datasets over the base video diffusion models **without additional training**, as shown in Table 3.
>
> FVD measures the alignment between the generated videos and reference videos, which in turn reflects both video-text alignment and video quality. Two factors together contributed to our improvements in the FVD score:
>
> 1. Our method, with an LLM processing input text prompts to generate dynamic scene layouts, has improved ability to understand the input text prompt. This is reflected in the improved video-text alignment.
> 2. Although improving video quality is not our main goal, our method also improves the quality of generated videos by using the intermediate dynamic scene layouts for planning, which in turn leads to better attribute binding and spatial layouts that are more reasonable.
>
> **For example,** in the first row of Fig. 1 in our paper, baseline ModelScope confuses the semantics of the bear and the pikachu mentioned in the prompt and generates an object that looks neither like a bear or like a pikachu. By giving spatial-temporal guidance to the video diffusion model, our method can successfully generate a bear and a pikachu with high visual quality, as required by the prompt. The ability for high-quality video generation in cases like this leads to our improved FVD score.
>
> > Is LLM able to generate reliable layouts using text without direction information, such as “a walking dog”.
>
> That is a great point! **As your requested, we ask the LLM to generate a walking dog without specifying the direction information, as shown in Figure 8(c\) in the appendix of the updated paper PDF.** We observe that the LLM is not only able to generate dynamic scene layouts for a walking dog but also **provide assumptions that it makes about the walking directions in the reasoning statement**. This shows that our LLM is indeed able to make reasonable guesses for ambiguous user prompts.
>
> ```
> Reasoning: A dog walking would have a smooth, continuous motion across the frame. **Assuming it is walking from left to right**, its x-coordinate should gradually increase. The dog's y-coordinate should remain relatively consistent, with minor fluctuations to represent the natural motion of walking. The size of the bounding box should remain constant since the dog's size doesn't change.
> ```
>
> If the assumption does not meet the user's expectation (e.g., the user wants a dog walking from the right instead), the user can reply to the LLM, and the LLM can incorporate additional information to satisfy the user's needs.
>
> Given the successful generation of the dynamic scene layouts, our layout-conditioned video generation method also successfully generates a video of a walking dog. We have added this example in the appendix of our updated manuscript. **Please see Sec A.1 and Fig. 8 (a) in our updated PDF for the video generation.** Thanks again for the suggestion!
>
> In light of this, we would like to kindly ask if there is anything to explain to make you consider raising your score. Additionally, we are open to addressing any additional concerns or answering any questions you may have!

---

> > ### Author Response · Authors · 2023-11-21
> > **Gentle reminder - 2 days left for the author-reviewer discussion**
> >
> > Dear reviewer ZTWf,
> >
> > We wanted to ask if you had a chance to check our response to your rebuttal questions regarding contribution and novelty, video quality improvement, and ambiguous prompts.
> >
> > Please let us know if we addressed your comments and/or if any further information or experiments would be helpful. We would be happy to provide them.
> >
> > Many thanks!
> > Authors

---

> > ### Comment · Reviewer_ZTWf · 2023-11-22
> > **Thanks for your response**
> >
> > After reading the authors' response, most of my concerns are solved, including the difference from the previous methods, results of ambiguous input, and technical novelty. I will maintain my rating at 6.

---

> > > ### Author Response · Authors · 2023-11-22
> > >
> > > Dear reviewer ZTWf,
> > >
> > > We are very glad that we have solved your concerns, and we will incorporate all you have suggested in our final version. Thank you again for your positive review!
> > >
> > > Best Regards,
> > > Authors

---

> ### Author Response · Authors · 2023-11-22
> **Gentle reminder: one day left for the author-reviewer discussion**
>
> Dear ZTWf,
>
> We appreciate your positive feedback on our work! We’re reaching out to see if you've had the opportunity to review our responses to your concerns about the novelty and contribution of our work, improvements in video quality, and clarification of ambiguous prompts. **We've conducted additional experiments and generated new videos based on your suggestions.**
>
> Could you kindly let us know if our responses have adequately addressed your comments? If there is any additional information or experiments that you believe would be beneficial, please don't hesitate to let us know. We're more than willing to provide further details during the author-reviewer discussion.
>
> Thank you so much!
>
> Authors

---

### Official Review · Reviewer_34cZ · 2023-11-01

**Soundness:** 3 good
**Presentation:** 3 good
**Contribution:** 3 good
**Rating:** 6
**Confidence:** 3

**Summary:**

Grounded Text-to-image generation has been studied by several papers recently. However, text-to-video geneartion with layout control is still unexplored. This paper tackles this task by proposing a training-free method by adding layout information through adjusting the attention maps of the diffusion UNet. Speficically, this paper first utilizes LLMs (GPT-4) to generate a multi-frame object layouts, then designs a layout-grounded video generator that encourages the cross-attention map to concentrate more on the bounding box areas. Extensive experiments for spatiotemporal dynamics evaluation have demonstrated the effectiveness of the proposed method.

**Strengths:**

- The paper is clearly written and easy to follow.

- The proposed method is training-free, which avoid the need for costly training with image/video data.

- Using LLM-generated layouts for videos is relatively unexplored. And it's natural to use the knowledge embedded in LLMs to general layouts for downstream video generation.

**Weaknesses:**

- Even though the proposed method is training free, it takes longer time during inference to generate videos due to the optimization steps needed for the energy function.

- Training-free layout control already exists in previous literatures [1, 2]. Therefore, the design of the energy function and backward guidance is not that novel.

- Ablation study of the model design is not given (e.g., number of DSL guidance steps, energy function design).


[1] Chen, Minghao, Iro Laina, and Andrea Vedaldi. "Training-free layout control with cross-attention guidance." arXiv preprint arXiv:2304.03373 (2023)

[2] Lian, Long, et al. "LLM-grounded Diffusion: Enhancing Prompt Understanding of Text-to-Image Diffusion Models with Large Language Models." arXiv e-prints (2023): arXiv-2305

**Questions:**

- Could the authors provide some reasoning why they report video-text similarity metric in Bain et al., 2021? It would be nice to also report CLIPScore, since its widely reported in other text-to-video generation baseilnes.

- The examples provided in the paper are with non-overlapping bounding boxes. Will the proposed method work well with overlapping layouts?

- If there are multiple objects, is the final energy function summing over the energy function corresponding to each object?

- It seems that the background areas of the generated images with proposed method are quite static (Fig1, 7, 8, 9). Is this because the model encourages static background, or becuase the visualized images happens to have relatively static background?

- Based on my understanding, another concurrent work, VideoDirectorGPT [1], is also for text-to-video generation with layout guidance. Even though the technical routes are different from this paper, it would be nice to have some discussions and comparison in the related work section.

[1] Lin, Han, et al. "Videodirectorgpt: Consistent multi-scene video generation via llm-guided planning." arXiv preprint arXiv:2309.15091 (2023)

**Details Of Ethics Concerns:**

No ethics concerns.

---

> ### Author Response · Authors · 2023-11-19
>
> We really appreciate your encouraging review and suggestions! Here we address each point of your comments:
>
> > Even though the proposed method is training free, it takes longer time during inference to generate videos due to the optimization steps needed for the energy function.
>
> Thanks for the great point! All previous methods based on backward guidance for training-free guidance (e.g., LMD, BoxDiff, Layout Guidance, Self-Guidance, and our method) take more inference time. The reason is that such methods need to find a way to leverage the knowledge inside the pretrained model during inference, without supervision from any external box-video pairs.
>
> However, training-free methods not only are free of training costs (time and compute requirements) but also have **no requirements on external data or human annotations (e.g., box annotations on frames)**. For example, our method can be applied off-the-shelf on video diffusion models such as ModelScope without external annotations. Moreover, training-free allows exploring a wide range of formulations without training them each time.
>
> Furthermore, even though inference time is usually not the main focus of this line of work (e.g., LMD, LayoutGPT, BoxDiff, etc.), **if low inference time is needed for the production environment, our method can be distilled into a conditioned generation model** so that the inference-time optimization of energy function is no longer required. Specifically, the generation of our method can potentially be distilled into a model with additional input tokens (e.g., ReCo) or attention adapters (e.g., GLIGEN), thus achieving fast generation without external datasets or additional human annotations. **This way, we get the advantages of being both free of external annotated data and having fast inference, enjoying the benefits of existing training-free and training-based methods.** We leave the distillation to future work as this is a general enhancement applicable to many training-free guidance methods.
>
> > Training-free layout control already exists in previous literatures [1, 2]. Therefore, the design of the energy function and backward guidance is not that novel.
>
> The energy function proposed in our pre-rebuttal manuscript, which applies spatial control per frame, is indeed a little similar to the energy function used in previous works. However, **as explained in the General Rebuttal \[Q1\],** we proposed a temporal energy term that provides guidance on the positions and velocities of center-of-mass of each object, which significantly improves our generation accuracy, especially on the temporal tasks. Since previous layout-to-image methods only focus on generating one image (i.e., do not take temporal consistency into accounts), **we believe our introduction of temporal center-of-mass energy term sets our method further apart from previous image layout conditioning methods mentioned by the reviewer.** We will update the manuscript accordingly. Thanks for the good point!
>
> > Ablation study of the model design is not given (e.g., number of DSL guidance steps, energy function design).
>
> Thank you for the suggestion. We have conducted ablation studies on the CoM energy function design as well as the number of DSL guidance steps in LVD. **Please refer to the General Rebuttal [Question 4] for more details.**

---

> ### Author Response · Authors · 2023-11-19
>
> > Could the authors provide some reasoning why they report video-text similarity metric in Bain et al., 2021? It would be nice to also report CLIPScore, since its widely reported in other text-to-video generation baseilnes.
>
> Good question! CLIPSIM, the video-text alignment score obtained from CLIP, is computed per-frame and then averaged across the feames ([reference](https://arxiv.org/abs/2104.14806)). This leads to its temporal ambiguity. **For example,** it could not distinguish a barrel floating from the left to the right vs from the right to the left. Instead, we use Frozen-In-Time, a CLIP-like model that takes in the whole video sequence and outputs a video-text alignment, which does not treat a video as a bag of frames and can potentially model object motion in the alignment score.
>
> **Per your request, we also evaluate our model and report the CLIPSIM metric compared to the baseline**, following a similar setting as Tab. 4 (Tab. 5 in the updated PDF) except we change the alignment model to a per-frame CLIP.
>
> |   | ModelScope | LVD on ModelScope |
> |:---:|:---:|:---:|
> | CLIPSIM | 0.2947 | **0.3001** |
>
> Our method gives better CLIPSIM compared to baseline ModelScope, indicating better text-to-video alignment.
>
> However, we would like to mention that previous benchmarks, such as FVD or similarity-based metrics (e.g., metrics based on CLIP or Bain et al., 2021), do not perform detailed checks for correctness in aspects such as spatial dynamics and generative numeracy. While these methods mainly focus on the semantic alignment and generation quality, **our proposed benchmark in Tab. 2 pioneers in quantitatively assessing the correctness in prompt understanding in video generation, opening up a direction to explore future video generation research.**
>
>
> > The examples provided in the paper are with non-overlapping bounding boxes. Will the proposed method work well with overlapping layouts?
>
>
> Yes. **We have added an example with overlapping layouts in Figure 8(b) in the paper appendix (Section A.1 in our updated PDF). Generally we find that the LVD also works with overlapping boxes.** Our energy function, with its `topk` operator, simply selects the most appropriate place to encourage object appearance, and two layout boxes can select different places due to the different topk selection. Therefore, two or more overlapping boxes will not conflict with each other.
>
> > If there are multiple objects, is the final energy function summing over the energy function corresponding to each object?
>
> You are right! The final energy function is the sum over the one for each object. We will make it clear in the manuscript.
>
> > It seems that the background areas of the generated images with proposed method are quite static (Fig1, 7, 8, 9). Is this because the model encourages static background, or becuase the visualized images happens to have relatively static background?
>
>
> Good point! Empirically we found that our base model, ModelScope, often tends to generate videos with small background variations, potentially because it uses a lot of static images in training. **To ensure that our model retains the ability to generate non-static background, we let it generate `a bear taking a shower in a lake, lightning strikes` as a simple example, which we show in Figure 8(c\) in the paper appendix (Sec A.1 in the updated PDF).** Although the background motion is not extreme, the model indeed generates rain dropping on the water and the lightning, showing that our method does not impede the ability to generate non-static background.
>
> > Based on my understanding, another concurrent work, VideoDirectorGPT [1], is also for text-to-video generation with layout guidance. Even though the technical routes are different from this paper, it would be nice to have some discussions and comparison in the related work section.
>
> Thanks for the suggestion! **We have included a reference to [1] and updated a comparison in the related work section in the paper.** Since this work was released on arXiv only a few days before the submission deadline, **[1] is a concurrent work that should not be used to devalue our work**.
>
> Our work has a different focus compared to [1]:
> 1. Our work is training-free and focuses on inference-time algorithms, whereas [1] trains adapters with human annotated images.
> 2. Our work focuses more on in-depth analysis on the capabilities of LLMs to generalize to unseen properties and to use knowledge in the weights rather than mimick in-context examples.
>
> **In light of the completion of several requested experiments and comparisons, we would like to kindly ask if there are any other concerns or additional questions that we can respond to or if you are willing to consider increasing your score!**

---

> ### Comment · Reviewer_34cZ · 2023-11-20
>
> Really appreciate for your detailed explanations to address my concern about CLIPSIM and overlaping bounding boxes!
>
> But I still have concern about the reasoning of static background you mentioned above. ModelScope is also trained on webvid dataset, and a lot of videos in webvid have obvious camera motions. So I'm not very convinced by the statement "ModelScope often tends to generate videos with small background variations".
>
> In the additional example you generated (Fig 8(c)), it seems that the background is still very static, even though there are lightings. But the lightings can be regarded as "foreground" objects rather than the background. Therefore, it would be nice if the authors can provide some examples with obvious background motion.
>
> And another minor question is about the resolution of videos generated from ModelScopeT2V. As described in their technical report, their model is trained and evaluated on resolution of 256\*256. In LVD, the videos are of resolution 512\*512. Do the authors observe some negative visual quality change after increasing the resolution?

---

> > ### Author Response · Authors · 2023-11-21
> > **Thank you for the response!**
> >
> > Thank you for the response! In the following we provide additional visualization and explanations regarding to your questions. Please feel free to let us know if these address your concerns.
> >
> > > ... I'm not very convinced by the statement "ModelScope often tends to generate videos with small background variations" ... Therefore, it would be nice if the authors can provide some examples with obvious background motion.
> >
> > Sorry for the confusion. We additionally generated a few example videos with background motion as requested:
> >
> > 1. [A car driving at night](https://anonymous.4open.science/r/ICLR2024_Submission2307-B317/car_driving_at_night.gif).
> > 2. [A car driving in rain](https://anonymous.4open.science/r/ICLR2024_Submission2307-B317/car_driving__in_rain.gif).
> > 3. [A man running at park](https://anonymous.4open.science/r/ICLR2024_Submission2307-B317/man_running.gif).
> >
> > In these videos, the camera is moving along with the foreground object, which causes the background motion in the video. This shows that our method does have the ability to generate videos with moving background.
> >
> > > And another minor question is about the resolution of videos generated from ModelScopeT2V. As described in their technical report, their model is trained and evaluated on resolution of 256*256. In LVD, the videos are of resolution 512*512. Do the authors observe some negative visual quality change after increasing the resolution?
> >
> > Thanks for the good point! We did not observe significant quality change between 256x256 and 512x512 videos, even though the ModelScopeT2V is trained on 256x256 videos. We plan to run evaluations also on 256x256 videos generated by ModelScopeT2V to investgate the effect of the inference resolution on generation accuracy for the next version of our work. For zeroscope, where we generated horizontal videos (e.g., in Fig. 1), the resolution is 576x320, which is the same as the resolution for training videos.
> >
> > **We would like to note that since our method is training-free, the resolution of the videos generated by our method can be easily switched by simply changing the method config, which suggests not devaluing our method based on this.**

---

> ### Author Response · Authors · 2023-11-22
>
> Dear reviewer 34cZ,
>
> We appreciate your positive and insightful feedback on our work! We would like to check again to see if you've had the opportunity to review our responses to your additional questions. We've conducted additional experiments to generate videos with background motion that arises from camera motion according to your request.
>
> **In light of our updates to our work addressing your concerns, completion of many requested experiments, and discussions about related work mentioned in the review, we would like to ask if you are willing to reconsider your score, and also if there are any additional questions that we can respond to during the last day of the author-reviewer discussion period!**
>
> Thank you so much!
>
> Best Regards,
>
> Authors

---

> > ### Comment · Reviewer_34cZ · 2023-11-23
> >
> > Dear authors,
> >
> > The additional video examples you provided addressed my concern about static background, so thanks again for your detailed explanation and additional efforts!
> >
> > A last question just out of curiosity: I noticed that the foreground objects in these new video examples are quite static, which means that the layouts (bounding boxes) generated by LLMs probably do not have large moments. Will this make LVD easier to generate videos with better temporal coherence? And have the authors observed a decrease in temporal coherence across frames in videos when there are significant layout changes?

---

> > > ### Author Response · Authors · 2023-11-23
> > >
> > > Thank you for acknowledging our efforts and results! To respond to your question, LVD can still preserve visual consistency when generating videos with both object (foreground) and camera (background) motion (i.e., it is not the case that LVD suffers from temporal incoherency when both foreground and background are moving).
> > >
> > > An example is shown [here](https://anonymous.4open.science/r/ICLR2024_Submission2307-B317/man_running_w_foreground_motion.gif). The man is running, with both the position and the size of the man changing significantly. The camera is also moving, causing background motion. The generation of the running man is still temporally coherent, leading to no flickering or abrupt changes across the frames.
> > >
> > > As you suggested, there could still be temporal inconsistencies in the videos that contain large foreground motion. A failure case is [here](https://anonymous.4open.science/r/ICLR2024_Submission2307-B317/dog_jumping.gif). The dog's face is slightly temporally inconsistent across the frames. However, since such inconsistencies also appear in ModelScope, it is partially due to the limitation of the base diffusion model itself and thus not necessarily a problem arising from LVD's design. We leave it to future exploration to address this problem for general text-to-video generation.
> > >
> > > In light of our clarifications and completion of many requested experiments, we would like to ask if you are willing to increase your score, and we are more than happy to answer any additional questions!

---

### Official Review · Reviewer_6SgA · 2023-11-01

**Soundness:** 2 fair
**Presentation:** 3 good
**Contribution:** 2 fair
**Rating:** 6
**Confidence:** 5

**Summary:**

The paper introduces a novel approach to text-conditioned video generation that seeks to address the limitations of current models, which struggle with complex spatiotemporal prompts and often produce videos with restricted or incorrect motion patterns. The key contribution is the LLM-grounded Video Diffusion (LVD) model that separates the video generation task into two steps: (1) using a Large Language Model (LLM) to generate dynamic scene layouts (DSLs) from text inputs, and (2) using these layouts to guide a diffusion model in generating the video. The approach is described as training-free and can be integrated with existing video diffusion models that allow for classifier guidance. Moreover, they introduce a benchmark for evaluating the alignment between input prompts and generated videos.

**Strengths:**

- The proposal of a training-free approach presents a pipeline that is well-suited for the application of off-the-shelf LLMs and diffusion models. Its simplicity yet effectiveness stands out as a notable strength.
- The discovery that LLMs can generate spatiotemporal layouts from text with only a limited number of in-context examples is noteworthy. It highlights the potential for a straightforward integration of LLM reasoning into text-to-video tasks.

**Weaknesses:**

- The idea of guidance via energy functions and cross-attention maps seems to be basically derived from BoxDiff (Xie et al., 2023;) and Chen et al. 2023a;. It is unclear how much of this work is based on previous research and how much is new. Since they are dealing with video generation using layouts, it would have been nice to see the authors' contribution in extending to the temporal axis, but this is not evident, which is disappointing.
- I am concerned that the scale of the sample size for the proposed DSL benchmark may be too small to conduct a sufficiently robust evaluation.
- The paper's contribution appears to lack novelty. There is existing work in text-to-image generation that has already established the capability of LLMs to create layouts, and this research seems to merely extend that to assess whether the same capability applies to temporal understanding. I didn't perceive any novel ideas stemming from the temporal aspect of the problem that would distinguish this work significantly from its predecessors.
- The paper seems to lack a detailed analysis or ablation studies concerning the prompts given to the LLM for generating Dynamic Scene Layouts (DSLs). Such investigations are crucial to understand how different prompts affect the LLM's output and the subsequent quality of the video generation. Further exploration in this area could significantly bolster the robustness of the presented approach.
- The paper's current framework could indeed benefit from additional ablation studies or analytical experiments to demonstrate the effectiveness of using DSLs for training-free guidance of text-to-video diffusion models. Moreover, a theoretical explanation of why this particular approach is effective would be valuable. It's important for the research to not only present the method but also to thoroughly validate and explain why certain choices were made and how they contribute to the overall performance and reliability of the model.

**Questions:**

- Can the authors elaborate on how the model performs with ambiguous or complex text prompts that might yield multiple valid interpretations in terms of spatial and temporal dynamics?
- Could the authors discuss any observed limitations or failure modes of the LVD approach, particularly in cases where the LLM might generate less accurate DSLs?
- (Minor point) Typographical error in Section 4, second paragraph. The sentence in question should indeed conclude with "feature map" instead of "feature ma." A revision is recommended for accuracy and clarity.

---

> ### Author Response · Authors · 2023-11-19
>
> Thanks for your detailed review and highly-valuable suggestions! **Accordingly, we improved our approach with a temporal energy function, added additional studies on a benchmark of a much larger scale, conducted key ablation studies to analyze our method in response to your comments.** We address each of your comments in detail:
>
> > Since they are dealing with video generation using layouts, it would have been nice to see the authors' contribution in extending to the temporal axis, but this is not evident, which is disappointing.
>
> Thanks for the suggestion! Indeed, although our method already greatly outperformed the baseline with the current energy function, innovation on temporal axis should further boost the temporal consistency of the generated objects. Therefore, **as introduced in the General Rebuttal \[Q1\]**, we added a temporal center-of-mass term, which significantly improves our generation accuracy by 10%, especially on the temporal dynamics and sequential movements, while reducing the mismatches between the generated objects and the dynamic scene layout input. **Our introduction of center-of-mass energy term sets our method further apart from previous image layout conditioning methods that do not deal with the temporal axis. We hope this novel energy term meets your expectation for novelty related to the temporal axis! Thanks for your suggestion again!**
>
> > I am concerned that the scale of the sample size for the proposed DSL benchmark may be too small to conduct a sufficiently robust evaluation.
>
> This point is totally valid! Taking your suggestions, we propose an enlarged version of our benchmark that is 10x larger compared to the previous version. Specifically, each task now has 5x more prompts in a similar structure as the previous version, with 500 prompts in total. For each prompt, we generate two videos from dynamic scene layout, with **1000 videos per benchmark run in total**.
>
> We run both the baseline and our method twice with the enlarged benchmark and **we observe evaluation results with small overall variation across the runs**.
>
> |   | Numeracy | Attribution | Visibility | Dynamics | Sequential | Average | **Variation** | Previous smaller benchmark |
> |:---:|:---:|:---:|:---:|:---:|:---:|:---:|---|---|
> | ModelScope Baseline (run 1/run 2) | 7.5%/5% | 65.5%/66% | 1.5%/1% | 18.5%/23.5% | 0%/0% | 18.6%/19.1% | **0.5%** | 16% |
> | Ours (run 1/run2) | 59.5%/55.5% | 94%/88.5% | 40%/50.5% | 65.5%/67.5% | 34%/32% | 58.6%/58.8% | **0.2%** | 58% |
>
> **Furthermore, our enlarged benchmark still results in very similar results compared to the previous smaller benchmark, which also validates the findings obtained with the previous benchmark. We hope the robustness in our benchmark results can address your concerns!**
>
> Finally, we would like to note that while previous benchmarks (e.g., CLIPSim and FVD) focus on generation quality and semantics, **our benchmark is a pioneer for correctness in spatial-temporal alignment in the literature,** which is of great significance for future research, especially given that the results justify the robustness of our benchmark.
>
> For the final version of the paper, we plan to run all our ablation studies on this enlarged benchmark and include the results in our work.

---

> ### Author Response · Authors · 2023-11-19
>
> > The paper's contribution appears to lack novelty. There is existing work in text-to-image generation that has already established the capability of LLMs to create layouts, and this research seems to merely extend that to assess whether the same capability applies to temporal understanding. I didn't perceive any novel ideas stemming from the temporal aspect of the problem that would distinguish this work significantly from its predecessors.
>
> Thanks for the comment. There may be some misconceptions on the capabilities justified by previous works that use LLMs for image layout generation. **There are two noteworthy distinctions between our work and previous works, which we briefly explain here, and we provide more detailed clarifications in the General Rebuttal \[Q2\]**.
> 1. Most of text-to-image-layout generation tasks introduced in previous works (LayoutGPT and LMD) can be completed without understanding the properties of the objects in the prompt in depth. In contrast, even a simple generation of dynamic scene layouts of "a falling ball" requires understanding of complex properties of the physical world, such as physical property (e.g., gravity which decides the trajectory of the ball), object interaction (e.g., ball hitting the ground), object properties (e.g., elasticity of ball which decides how much the ball will bounce), and camera properties (e.g., perspective geometry which decides if the size of the ball will change based on its distance to the camera), which are all **highly non-trivial for an LLM and have not been assessed, demonstrated, or evaluated by previous works**. **In contrast, we show that LLMs demonstrate the understanding of these properties, as illustrated in Figure 4 in the paper.**
> 2. Previous works either only evaluate properties that are **already seen in in-context examples** (e.g., the 4 main tasks that LMD showcases are already demonstrated in the in-context examples in Table 6 of [the LMD paper](https://arxiv.org/pdf/2305.13655.pdf)), or **retrieve related layouts from a layout database** as in-context examples (LayoutGPT). **Neither work demonstrates whether LLMs have the knowledge for generalization without relevant in-context examples.** In contrast, in Section 3 (Discoveries part), we show that LLMs can generalize to physical properties or object properties that are **unseen in the in-context examples** (e.g., as shown in Figure 4 (c\) in the paper, when generating layouts of throwing a paper airplane, it can figure out the airplane will slide in the air due to its weight and aerodynamics, while similar concepts are never mentioned in the in-context examples). This implies LLMs have an intrinsic understanding of the physical world instead of just emulating from in-context examples.
>
> We will clarify these two points further in our manuscript. Thanks again for the valuable feedback!
>
> > The paper seems to lack a detailed analysis or ablation studies concerning the prompts given to the LLM for generating Dynamic Scene Layouts (DSLs). Such investigations are crucial to understand how different prompts affect the LLM's output and the subsequent quality of the video generation. Further exploration in this area could significantly bolster the robustness of the presented approach.
>
> Thank you for the suggestion! We have added ablation studies on how the number, content, and format of prompted in-context examples will affect the generation quality. **Please refer to the General Rebuttal \[Q3\],** in which we show that 3 in-context examples already result in high-quality dynamic scene layout generation and LLM is robust to different in-context examples in both content and format.
>
> > The paper's current framework could indeed benefit from additional ablation studies or analytical experiments to demonstrate the effectiveness of using DSLs for training-free guidance of text-to-video diffusion models.
>
> Thank you for the comment. To analyze different design choices of the DSL-to-video stage in LVD, we conduct ablation studies on the CoM energy function design as well as the number of guidance steps in the DSL-grounded video generation pipieline. **Please refer to the General Rebuttal \[Q4\] for more details.**

---

> ### Author Response · Authors · 2023-11-19
>
> > Can the authors elaborate on how the model performs with ambiguous or complex text prompts that might yield multiple valid interpretations in terms of spatial and temporal dynamics?
>
> That is a good point! The potential ambiguity in the user-provided prompts is indeed worth exploring. We find that LLMs can oftentimes perform reasonable guesses if the spatial-temporal information is missing in the user-given prompt.
>
> For example, with a simple prompt as `a walking dog`, the LLM gives the following reasoning statement, with the assumption clearly written in it:
> ```
> Reasoning: A dog walking would have a smooth, continuous motion across the frame. **Assuming it is walking from left to right**, its x-coordinate should gradually increase. The dog's y-coordinate should remain relatively consistent, with minor fluctuations to represent the natural motion of walking. The size of the bounding box should remain constant since the dog's size doesn't change.
> ```
>
> This allows the subsequent layout generation to be consistent with respect to the assumption of walking from the left to the right. **We show the generated video of this example in Figure 8(a) in appendix of the updated paper PDF.**
>
>
> Additionally, if the assumption does not meet the user's expectation (e.g., the user wants a dog walking from the right instead), the user can reply to the LLM, and the LLM will incorporate additional information to satisfy the user's needs.
>
> > Could the authors discuss any observed limitations or failure modes of the LVD approach, particularly in cases where the LLM might generate less accurate DSLs?
>
> Thank you for the suggestion. We show two failure cases in the DSL-to-video stage in Figure 10 and Figure 11 of the updated paper PDF. For the text-to-DSL stage, although the LLM-generated DSLs have a high accuracy in following the prompt (98% accuracy in Table 1 in the paper), it still has some failure cases. The failures mainly come from generating complex dynamics such as sequential movements. One failure case, which we added in the updated paper PDF, is shown as below and also demonstrated in Figure 9 in Section A.6 of the paper appendix:
>
> *prompt*: A realistic lively video of a top-down viewed scene in which a ball initially on the lower left of the scene. It first moves to the upper left of the scene and then moves to the upper right of the scene, outdoor background.
>
> *generated boxes* (visualized in Figure 9 of the updated paper PDF):
> ```
> Frame 0: (0.097, 0.878, 0.195, 0.976)
> Frame 1: (0.097, 0.683, 0.195, 0.781)
> Frame 2: (0.097, 0.488, 0.195, 0.585)
> Frame 3: (0.292, 0.488, 0.390, 0.585)
> Frame 4: (0.488, 0.488, 0.585, 0.585)
> Frame 5: (0.683, 0.488, 0.781, 0.585)
> ```
>
> Note that for each frame, the box is represented by 4 float numbers, where the first two are the x and y coordinates of the upper left corner of the box, and the last two are the x and y coordinates of the lower right corner of the box.
>
> In this case, although the prompt indicates the ball moves to the upper left corner of the scene, the LLM does not generate the box in the upper left corner, but instead a box slightly above the middle of the whole scene. The failure is probably because the sequential movements contain multiple stages and LLM has a higher probability to oversee or misunderstand one of the stages. We will add these discussions on the failure cases to the paper.
>
>
> > (Minor point) Thank you for pointing out this typographical error in Section 4, second paragraph. The sentence in question should indeed conclude with "feature map" instead of "feature ma." A revision is recommended for accuracy and clarity.
>
> Thanks for the good point! We have fixed this in our manuscript.
>
> **In light of the additional improvements to our methods and benchmarks we've provided, as well as the completion of numerous requested experiments, we would like to kindly ask if you might be open to increasing your assessment score, and if there are any new concerns or further questions that we can address for you!**

---

> > ### Author Response · Authors · 2023-11-21
> > **Gentle reminder - 2 days left for the author-reviewer discussion**
> >
> > Dear reviewer 6SgA,
> >
> > We wanted to ask if you had a chance to check our response to your rebuttal questions regarding contribution and novelty, benchmark size, ablation studies, and failure cases.
> >
> > Please let us know if we addressed your comments and/or if any further information or experiments would be helpful. We would be happy to provide them.
> >
> > Many thanks!
> > Authors

---

> > > ### Comment · Reviewer_6SgA · 2023-11-21
> > >
> > > Thank you for the thorough responses to the concerns I raised. All issues have been resolved. The strengthened benchmark contributions through our discussions are particularly noteworthy and hold potential impact for the field. Consequently, I am adjusting my score towards an accept.

---

> > > > ### Author Response · Authors · 2023-11-21
> > > >
> > > > Thanks for checking our rebuttal and providing insightful suggestions! We really appreciate all the points you've mentioned and will incorporate the new results into our final version.

---

### Author Response · Authors · 2023-11-19
**General Rebuttal**

We would like to thank all the reviewers for their thoughtful reviews and encouraging feedback. We are especially glad that the reviewers believe that our method's "simplicity yet effectiveness stands out as a notable strength" (6SgA), "using LLM-generated layouts for videos is relatively unexplored" (34cZ), and our finding "interesting", exploration "inspiring", and results "promising" (ZTWf).

We would also like to thank the reviewers for all the insightful suggestions. **We present several improvements to our work as well as additional key experimental results that we believe significantly strengthen our work.** We address the common questions from the reviewers in this general rebuttal.

---

> ### Author Response · Authors · 2023-11-19
>
> ## \[Q1\] Differences with previous layout-conditioned image generation methods / Adding guidance to the temporal axis
>
> One common concern (Reviewer 6SgA, 34cZ, ZTWf) is the lack of differences with previous layout-conditioned image generation methods such as Layout Guidance (Chen et al. 2023a) and BoxDiff (Xie et al., 2023) since the pre-rebuttal version of our work only leverages a spatial energy term applied to each frame. **To resolve this concern, we added an additional temporal energy function to specifically provide guidance across the temporal axis (i.e., across the frames).**
>
> To extend the current LVD energy function to the temporal axis and utilize the temporal information, we propose an additional CoM (Center of Mass) energy term which promotes the position and motion consistency between temporal dynamics of generated objects and layouts, **which leads to benefits especially for tasks that require strong temporal understanding abilities.**
>
> Specifically, the proposed energy function first calculates the CoM of the cross-attention map as well as the bounding box mask, and then penalizes the difference between the position and velocity of two CoMs. In this way, the generated object can better follow the spatial position and temporal dynamics of the generated boxes.
>
> In the table below, we report the performance of LVD with CoM energy term, and compare to vanilla LVD without CoM energy term. To further benchmark the alignment between the generated objects and the input scene layout, we also add an additional metric which measures the average distance between the center of each generated object and its corresponding input bounding box, termed "Generation Mismatch".
>
> **We can see that the CoM energy term, as a temporal term, significantly improves LVD's performance, especially for object dynamics and sequential movements that are at the core of text-to-video generation.** It also generates objects that better follow the layouts, as measured by box distance. The CoM energy function improves the average accuracy by 10% over the vanilla LVD and greatly reduces the generation mismatch. **We have updated our paper PDF to include this new energy term in our method section.**
>
> |   | Numeracy Accuracy | Attribution Accuracy | Visibility Accuracy | Dynamics Accuracy | Sequential Accuracy | Overall Accuracy (↑)  |   | Generation Mismatch (↓) |
> |:---:|:---:|:---:|:---:|:---:|:---:|:---:|:---:|:---:|
> | LVD w/o CoM energy function | **45%** | **100%** | 50% | 60% | 35% | 58% |   | 0.38 |
> | LVD w/ CoM energy function | 40% | **100%** | **55%** | **80%** | **65%** | **68%** |   | **0.29** |
>
> The intuition behind this is that the `topk` function used by both our original energy term and previous methods mostly cares about whether an object is generated somewhere in the box. However, the exact position is loosely constrained.
>
> **This is usually not a problem for image generation**, since a slight deviation for object positions within an image is usually tolerable. However, in a video generation setting, the `topk` function may ask the model to generate an object on different areas of the input box in different frames, especially during the first few denoising steps when the cross-attention maps are noisy and are only loosely-correlated across the temporal axis. This could mislead the video diffusion model to believe the two objects in different frames are disjoint, thus applying the wrong temporal prior, making the energy function difficult to optimize, and finally causing misalignment with the input text prompt. In contrast, **our center-of-mass energy term applies a precise control of both object positions and velocities, which is vital for the video diffusion model to correctly apply its temporal prior to generate videos that align with the layout boxes**, thereby improving the alignment between the input condition and the actual generation.
>
> Finally, we implement the energy functions from previous text-to-image work Layout Guidance [1] and BoxDiff [2] applied on the same video diffusion model for fair comparison. We observe that our formulations, especially with the CoM energy term, greatly outperform previous works.
>
> |   | Layout Guidance | BoxDiff | LVD w/o CoM (Ours) | LVD w/ CoM (Ours) |
> |:---:|:---:|:---:|:---:|:---:|
> | Overall Accuracy (↑) | 44% | 45% | 58% | **68%** |
>
> Our center-of-mass energy term deals with the guidance across the temporal axis and significantly improves the generation alignment, which **adds novelty to our work and further distinguishes our work from previous layout-conditioned text-to-image methods that do not deal with the temporal dimension**.

---

> > ### Author Response · Authors · 2023-11-19
> >
> > ## \[Q2\] Significance and novelty of our findings for LLMs' generalization in dynamic scene layout generation / Differences with previous observations presented in works for LLM for text-to-image generation
> >
> > One common concern (Reviewer 6SgA, 34cZ, ZTWf) is that given previous observations that LLMs can generate spatial layouts for image generation, our finding that LLMs can generate spatial-temporal dynamic scene layouts for video generation is trivial. Here we would like to emphasize two points that distinguish us from previous work and add to the significance of our findings in this paper.
> >
> > ***First, generating spatial-temporal layouts requires the understanding of much more complex properties of the physical world, which is not shown or discussed in the previous work.***
> >
> > In fact, most of text-to-static-layout generation tasks introduced in previous works (LayoutGPT and LMD) can be completed without in-depth understanding of the properties of the objects in the prompt. **For example,** generating an image with 5 birds mostly entails drawing 5 boxes with similar sizes, which only requires limited understanding of the properties of a bird.
> >
> > However, video generation requires understanding highly non-trivial properties of the physical world, including physical properties, object motion trajectory, object interactions, and perspective geometry. **For example,** modeling a falling ball requires understanding physical properties (e.g., gravity that determines the curve and speed of falling), object interactions (e.g., the fact that the ball will hit the ground), object properties (e.g., elasticity that determines whether and how much the ball will bounce back), and camera properties (e.g., perspective geometry that determines whether the ball will be smaller due to potential distance change to the camera).
> >
> > Our analysis and assessment, presented in Figure 4 in the paper, demonstrates that LLMs indeed have this level of understanding even without relevant in-context examples, which was neither analyzed or even discussed in previous text-to-image related works. **None of the above-mentioned complexities are assessed, demonstrated, or evaluated in previous LLM for text-to-image generation works.**
> >
> > ***Second, we show a stronger generalization capability of LLMs that is not demonstrated in previous work.***
> >
> > Concretely, although previous works show LLMs’ understanding of properties of spatial layouts, they either only evaluate properties that are **already seen in in-context examples** (e.g., the 4 main tasks that LMD showcases are already demonstrated in the in-context examples in Table 6 of [the LMD paper](https://arxiv.org/pdf/2305.13655.pdf)), or **retrieve related layouts from a layout database** as in-context examples (LayoutGPT). **Neither work demonstrates whether LLMs have the knowledge for generalization without relevant in-context examples.**
> >
> > However, in Section 3 (Discoveries), **we show that LLMs can generalize to physical properties or object properties that are unseen in the in-context examples** (e.g., when throwing a paper airplane, it will slide in the air because of its weight and aerodynamics, which is never shown in the in-context examples). This implies LLMs have an intrinsic understanding of the physical world instead of just mimicking the in-context examples.
> >
> > This discovery is significant both for future research and applications because it not only shows **the potential generalization outside of the given examples without task-specific fine-tuning** but also suggests that **LLMs for dynamic scene layout generation only requires few in-context examples to work well**, which is especially useful for downstream applications. We will make these points clear in the paper.

---

> ### Author Response · Authors · 2023-11-19
>
> ## \[Q3\] Ablation studies on LLMs' capability of generating DSLs from text
>
> Reviewers (6SgA, 34cZ) are concerned that ablation studies on LLM's capability of generating DSLs from text are lacking. Especially, Reviewer 6SgA points out that an analysis of the prompts and in-context examples given to LLM is needed. **Here we conduct ablation studies on how the number, the content, and the format of in-context examples can affect LLM's capability of generating DSLs.** We will add the results in the paper.
>
> **We first test the effect of different numbers of in-context examples.** We test with 1, 3, and 5 in-context examples, respectively, and report the performance of each on our proposed benchmark. Results are shown below. We can see that using only 1 in-context prompts can degrade the average accuracy by 13%. This is likely because the diversity of in-context examples is not enough for sufficiently informing the LLM about the task formulation. However, we find that the performance of 3 and 5 examples are approximately the same. This indicates that 3 examples is enough to enable LLMs to generalize to many unseen test cases, implying that the capability of understanding the spatial and temporal properties of the physical world does not come from exhaustive in-context examples, but comes from the pretrained LLM itself. Therefore, by default we use 3 in-context examples in the paper. **We will add this ablation to the paper to strengthen our results.**
>
> |   | 1 Example | 3 Examples | 5 Examples |
> |:---:|:---:|:---:|:---:|
> | Overall Accuracy (↑) | 45% | **58%** | **58%** |
>
> **We also test if LVD is robust to different in-context examples.** We have a set of 5 in-context examples and randomly sample 3 examples to prompt LLM. We repeat this test for 3 times, and report the performance of each test. The results are shown below. We can see that the variation across different tests is about 2%,  which indicates the robustness of our method to different in-context examples. We will update the results in the paper.
>
> |   | test 1 | test 2 | test 3 |
> |:---:|:---:|:---:|:---:|
> | Overall Accuracy (↑) | 56% | 58% | 54% |
>
> Finally, **we test whether LVD is robust to different prompting formats for the in-context examples and outputs**. Specifically, rather than the using mostly natural language for representing the examples and outputs (in Table 8 of our work), we let the LLM process in-context examples in a JSON format and directly output a JSON response to represent the dynamic scene layouts. We run this ablation on our benchmark and show negligible variations of the generation accuracy. This indicates that our use of LLM for dynamic scene layout generation does not depend on a particular form of layout representation format in prompting.
>
> |   | Text Format | JSON Format |
> |:---:|:---:|:---:|
> | Overall Accuracy (↑) | 58% | 59% |
>
> ## \[Q4\] Ablation studies on the algorithm for generating videos from the DSLs
>
> Reviewers (6SgA, 34cZ) are concerned that ablation studies on DSL-grounded video generation algorithm are lacking. **To analyze the effect of different design choices in the algorithm, we conduct ablation studies on the energy function as well as the number of DSL guidance steps, respectively.** We will update the results in the paper.
>
> **First, to analyze the effect of the proposed CoM energy function**, we compare video generation results with or without this function. Detailed results are shown in General Rebuttal \[Q1\]. We find that the proposed energy function greatly boosts the performance of video generation, especially in temporal tasks, validating this design choice.
>
> **Second, we ablate on the number of DSL guidance steps between two denoising steps in LVD**. We test 1, 3, 5, and 7 steps of DSL guidance, respectively. The results are shown below. We can see that the average accuracy improves when adding more DSL guidance steps from 1 to 5 steps, and the performance plateaus when we use more than 5 steps. This means 5 steps is enough and probably the best choice considering the tradeoff between performance and efficiency. We will also add this result to our paper.
>
> |   | 1 step | 3 steps | 5 steps | 7 steps |
> |:---:|:---:|:---:|:---:|:---:|
> | Overall Accuracy (↑) | 33% | 49% | 58% | **59%** |

---

### Meta-Review · Area_Chair_E5ys · 2023-12-08

**Metareview:**

This paper presents a novel approach, LLM-grounded Video Diffusion (LVD), for text-conditioned video generation, introducing a training-free model that leverages Large Language Models (LLMs) for spatiotemporal layout generation. While the proposal exhibits simplicity and effectiveness, concerns arise about novelty and attribution. The paper acknowledges influences from recent works, prompting questions about the extent of its contribution. Evaluation benchmarks and ablation studies could be more comprehensive. Despite a clear presentation and promising results, additional validation and differentiation from existing methods are needed for a more impactful contribution. Addressing these concerns will enhance the paper's soundness and overall contribution.

All the reviewers were satisfied with the authors' response. Accept.

**Justification For Why Not Higher Score:**

The paper addresses the layout control by LLM for video generation, which is new. However, as compared to the recent SOTA of video generation, the quality cannot be considered as impressive.

**Justification For Why Not Lower Score:**

Nice paper and good implementation. All reviewers unanimously recommend accept. Overall, it is a promising contribution to the ICLR community.

---

### Decision · Program_Chairs · 2024-01-16

Accept (poster)